JCB Journal of Cell Biology

# α-Tubulin detyrosination impairs mitotic error correction by suppressing MCAK centromeric activity

Luísa T. Ferreira[1,2], Bernardo Orr[1,2], Girish Rajendraprasad[3], António J. Pereira[1,2], Carolina Lemos[2,4,5], Joana T. Lima[1,2], Clàudia Guasch Boldú[3], Jorge G. Ferreira[1,2,6], Marin Barisic[3,7], and Helder Maiato[1,2,6]

Incorrect kinetochore–microtubule attachments during mitosis can lead to chromosomal instability, a hallmark of human cancers. Mitotic error correction relies on the kinesin-13 MCAK, a microtubule depolymerase whose activity in vitro is suppressed by α-tubulin detyrosination—a posttranslational modification enriched on long-lived microtubules. However, whether and how MCAK activity required for mitotic error correction is regulated by α-tubulin detyrosination remains unknown. Here we found that detyrosinated α-tubulin accumulates on correct, more stable, kinetochore–microtubule attachments. Experimental manipulation of tubulin tyrosine ligase (TTL) or carboxypeptidase (Vasohibins-SVBP) activities to constitutively increase α-tubulin detyrosination near kinetochores compromised efficient error correction, without affecting overall kinetochore microtubule stability. Rescue experiments indicate that MCAK centromeric activity was required and sufficient to correct the mitotic errors caused by excessive α-tubulin detyrosination independently of its global impact on microtubule dynamics. Thus, microtubules are not just passive elements during mitotic error correction, and the extent of α-tubulin detyrosination allows centromeric MCAK to discriminate correct vs. incorrect kinetochore–microtubule attachments, thereby promoting mitotic fidelity.

## Introduction

Successful chromosome segregation during mitosis requires that each sister kinetochore (KT) is stably attached to microtubules (MTs) oriented to opposite spindle poles (amphitelic attachments). However, due to stochastic interactions between KTs and spindle MTs early in mitosis, many chromosomes establish erroneous attachments that have been implicated in chromosomal instability, a hallmark of human cancers (Bakhoum and Cantley, 2018; Cimini et al., 2003). To prevent this, cells rely on error correction mechanisms that regulate MT dynamics at the KT (Bakhoum et al., 2009). These mechanisms may act globally through the regulation of Cdk1 activity during early mitosis (Kabeche and Compton, 2013) or, more locally, by promoting MT detachment from KTs in response to low centromeric tension (Cimini et al., 2006; Liu et al., 2009). At the heart of this local error correction mechanism, the Aurora B kinase regulates the recruitment and/or activity of several centromeric/KT proteins, including the kinesin-13 MCAK (Andrews et al., 2004; Bakhoum et al., 2009; Knowlton et al., 2006; Lan et al., 2004), a potent MT depolymerase present in animal cells. Therefore, mitotic error correction is currently viewed as a "blind" process that results

from the nondiscriminatory renewal of MTs at the KT interface, regardless of their attachment status (i.e., correct or incorrect).

How MCAK mediates mitotic error correction has been difficult to determine, mostly due to its independent localization at centromeres/KTs and MT plus ends, and its global impact on spindle MT dynamics (Bakhoum et al., 2009; Domnitz et al., 2012; Huang et al., 2007; Kline-Smith et al., 2004; Rizk et al., 2009; Wordeman et al., 2007). Interestingly, the contribution of MCAK for KT MT (kMT) turnover appears to occur primarily in metaphase (Bakhoum et al., 2009), when most KT-MT attachments are amphitelic and stabilized, and it does so without any measurable impact on MT polymerization dynamics associated with MT poleward flux (Ganem et al., 2005). This apparent paradox led us to hypothesize that centromeric MCAK is able to discriminate correct and incorrect KT-MT attachments independently of its global effect on spindle MT dynamics. In support of this hypothesis, active MCAK is enriched at centromeres/KTs of misaligned chromosomes (Andrews et al., 2004; Lan et al., 2004), as well as in aligned chromosomes with erroneous merotelic attachments (Knowlton et al., 2006).

[1]Chromosome Instability & Dynamics Group, i3S - Instituto de Investigação e Inovação em Saúde, Universidade do Porto, Porto, Portugal;   [2]Instituto de Biologia Molecular e Celular, Universidade do Porto, Porto, Portugal;   [3]Cell Division and Cytoskeleton, Danish Cancer Society Research Center, Copenhagen, Denmark;   [4]UnIGENe, i3S - Instituto de Investigação e Inovação em Saúde, Universidade do Porto, Porto, Portugal;   [5]Instituto de Ciências Biomédicas Abel Salazar, Universidade do Porto, Porto, Portugal;   [6]Cell Division Group, Experimental Biology Unit, Department of Biomedicine, Faculdade de Medicina, Universidade do Porto, Porto, Portugal;   [7]Department of Cellular and Molecular Medicine, Faculty of Health Sciences, University of Copenhagen, Copenhagen, Denmark.

Correspondence to Helder Maiato: maiato@i3s.up.pt

In vitro reconstitution experiments have shown that MCAK activity is significantly suppressed by α-tubulin detyrosination (Peris et al., 2009; Sirajuddin et al., 2014), a posttranslational modification that accumulates on long-lived MTs (Nieuwenhuis and Brummelkamp, 2019). α-tubulin detyrosination has been recently implicated in mitosis and meiosis, neuronal processes and cognitive brain function, heart and skeletal muscle contraction, and cancer (Akera et al., 2017; Barisic et al., 2015; Chen et al., 2018; Erck et al., 2005; Kerr et al., 2015; Lafanechère et al., 1998; Liao et al., 2019; Pagnamenta et al., 2019; Robison et al., 2016). The detyrosination/tyrosination cycle involves the catalytic removal of the C-terminal tyrosine of most mammalian α-tubulin isoforms by tubulin carboxypeptidases, such as the recently identified Vasohibin (VASH) 1/VASH2-SVBP complexes (Aillaud et al., 2017; Nieuwenhuis et al., 2017), followed by retyrosination of soluble α-tubulin by tubulin tyrosine ligase (TTL; Ersfeld et al., 1993).

Here we combined powerful gene manipulation tools (including RNAi, small-molecule inhibition, protein overexpression, and CRISPR-Cas9 gene editing) with state-of-the-art microscopy, including a novel super-resolution microscopy technique (Pereira et al., 2019), to investigate whether MCAK activity required for mitotic error correction is regulated by α-tubulin detyrosination. Our findings support a physiological role for α-tubulin detyrosination in the discrimination between correct and incorrect KT-MT attachments, establishing a new paradigm in the control of mitotic fidelity.

## Results

### Detyrosinated α-tubulin accumulates on correct, more stable, KT-MT attachments

To investigate whether the mitotic error correction machinery "reads" the tyrosination/detyrosination state of α-tubulin on kMTs, we started by quantifying the ratio of detyrosinated/tyrosinated α-tubulin immediately adjacent to the KT in distinct experimental conditions that favor a particular attachment configuration. These included short-lived syntelic (when both KTs from a pair are attached to MTs from the same pole) or monotelic (when only one KT from the pair is attached to MTs) attachments, typically observed in monopolar spindles after kinesin-5 inhibition with monastrol (Kapoor et al., 2000); more stable amphitelic attachments that form during normal metaphase; and finally, hyperstabilized amphitelic attachments induced by treatment with 10 nM taxol for 1 h. We found that correct amphitelic attachments show higher α-tubulin detyrosination near KTs when compared with incorrect/incomplete syntelic/monotelic attachments, and this tendency correlated with increased MT stability (Fig. 1, A–F). Thus, α-tubulin detyrosination near KTs increases with the establishment of correct, more stable, MT attachments.

### Constitutively high α-tubulin detyrosination near KTs leads to mitotic errors

To investigate whether α-tubulin detyrosination impacts mitotic error correction, we inactivated TTL either by RNAi or CRISPR/Cas9-mediated gene knockout (KO) in human U2OS cells. Both conditions caused a significant increase in the overall α-tubulin detyrosination levels, with a proportional reduction in the tyrosinated α-tubulin and without detectable changes in the total tubulin pool (Figs. S1 A and S2 A). This was also the case in mitotic spindles, in which detyrosinated α-tubulin was increased along KT fibers, including in the immediate vicinity of the KT, and astral MTs after TTL inactivation (Fig. 2, A–C; Fig. S1, B and C; and Fig. S2 B). Live-cell imaging 72 h after TTL RNAi in cells stably expressing H2B-histone-GFP/mCherry-α-tubulin revealed a consistent and significant increase in the frequency of anaphase cells with lagging chromosomes (Fig. 2, D–F). Interestingly, this increase in chromosome missegregation gradually attenuated to control levels after chronic TTL inactivation (TTL KO), despite the continuous increase in detyrosinated α-tubulin over time (Fig. S2, C–F). This suggests the existence of compensatory or cellular adaptation mechanisms to the chronic loss of TTL, as shown previously for TTL KO mice (Erck et al., 2005).

To test whether the observed mitotic errors after TTL RNAi were a consequence of constitutively high α-tubulin detyrosination and not due to other putative roles of TTL, we performed rescue experiments with RNAi-resistant wild-type human TTL fused with YFP (TTL-YFP[R]) or a corresponding catalytic dead version (TTL[CD]-YFP[R]) in which a glutamic acid at position 331 was replaced by glutamine (E331Q), rendering TTL defective in α-tubulin retyrosination, but not tubulin binding (Prota et al., 2013; Szyk et al., 2011). Western blot and confocal fluorescence microscopy analyses confirmed the RNAi resistance of both TTL-YFP[R] and TTL[CD]-YFP[R] compared with conventional TTL-YFP, as well as the recovery of nearly normal α-tubulin detyrosination levels (total and in mitotic spindles) in the TTL-YFP[R], but not TTL[CD]-YFP[R] rescue conditions (Fig. S3, A and B). Surprisingly, we noticed that ectopic expression of any version of TTL-YFP resulted in the overexpression of endogenous TTL (Fig. S3 A). To exclude potential deleterious effects associated with the overexpression of ectopic and endogenous TTL (Szyk et al., 2011), we considered only cells expressing low-to-mild levels of TTL-YFP[R] or TTL[CD]-YFP[R] that retained an apparently normal spindle architecture in our quantification of mitotic errors. Accordingly, we found that TTL-YFP[R], but not TTL[CD]-YFP[R], was able to rescue the increased frequency of anaphase cells with lagging chromosomes observed upon TTL RNAi (Fig. S3 C). These results not only demonstrate the specificity of the TTL RNAi phenotype, but also directly indicate that the observed mitotic errors were dependent on the catalytic activity of TTL required for α-tubulin retyrosination.

Last, experimental increase of α-tubulin detyrosination levels independently of TTL by overexpression of the tubulin carboxypeptidase VASH1-SVBP led to an equivalent increase in anaphase cells with chromosome segregation errors when compared with TTL inactivation by RNAi (Fig. 3, A–C, F, and G). Conversely, RNAi-mediated depletion of VASH1 and VASH2 (with or without SVBP depletion) reduced chromosome missegregation events below control levels (Fig. 3, D–G). Overall, these data indicate that experimental modulation of α-tubulin detyrosination levels on mitotic spindles,

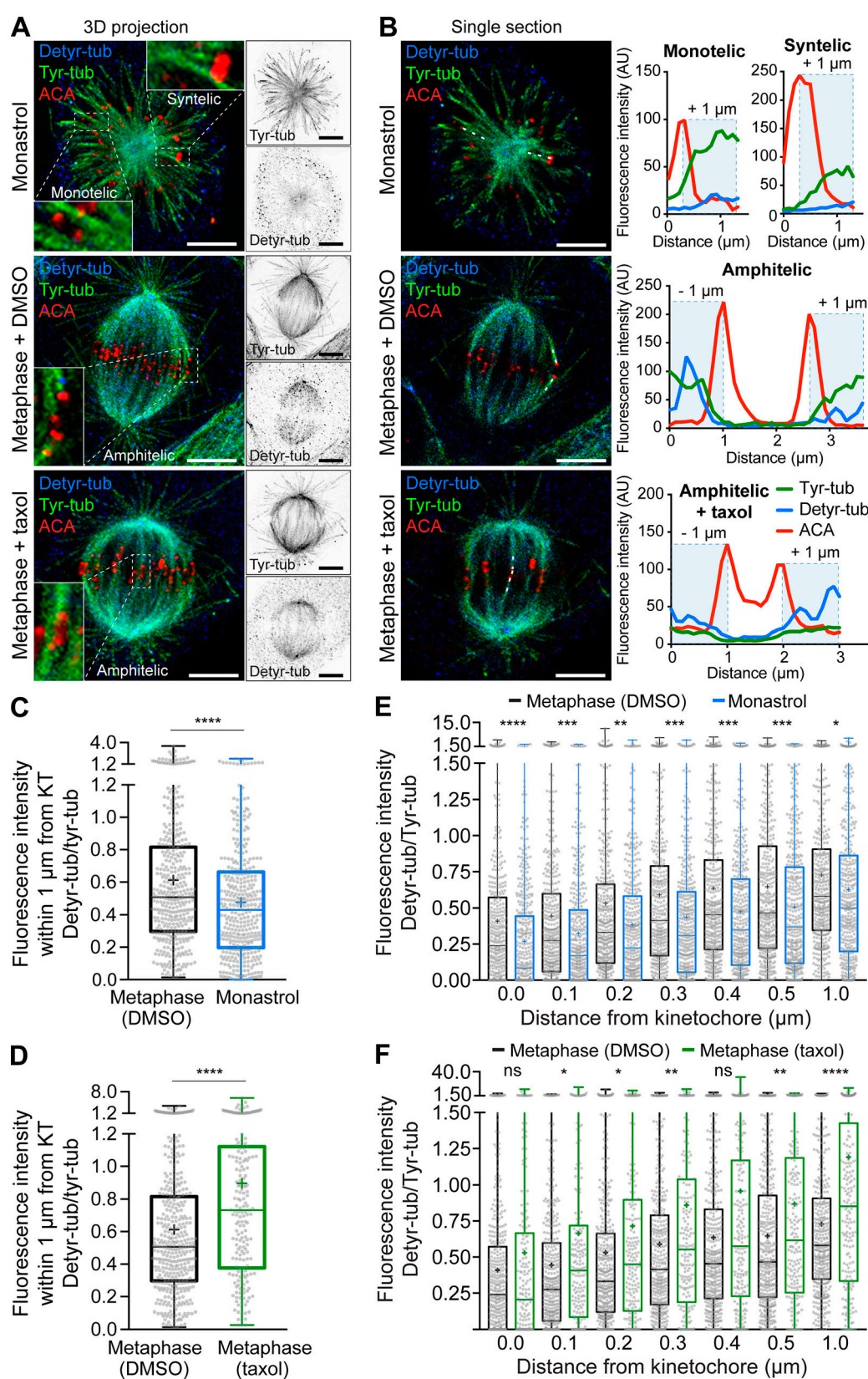

Figure 1. **MT detyrosination near KTs increases with the establishment of correct, more stable, MT attachments. (A)** Confocal analysis of U2OS cells by immunofluorescence for detyrosinated and tyrosinated α-tubulin (detyr-tub and tyr-tub, respectively), and KTs with ACAs. Maximum intensity 3D projection of representative cells for each attachment configuration (zoomed images highlight monotelic/syntelic attachments—monastrol, amphitelic attachments— DMSO and hyper-stable amphitelic attachments—taxol). **(B)** Single sections from the respective confocal series shown in A with dashed lines indicating line-scanned representative KT fibers (k-fibers). Scale bars, 5 μm. Line-scan plots of the selected KT fibers indicated in B. **(C and D)** Interquartile representation of

the mean cumulative fluorescence intensity ratio between detyrosinated/tyrosinated α-tubulin within 1 µm distance from the KT centroid in monotelic/syntelic attachments (monastrol: n = 391 k-fibers, ~20 k-fibers/cell, five cells/experiment, four independent experiments) and hyper-stable amphitelic attachments (taxol: n = 199 k-fibers, ~20 k-fibers/cell, five cells/experiment, two independent experiments), respectively, compared with untreated amphitelic (DMSO: n = 396 k-fibers, ~20 k-fibers/cell, five cells/experiment, four independent experiments) attachments. **(E and F)** Interquartile representation of the fluorescence intensity ratio between detyrosinated/tyrosinated α-tubulin as a function of distance from the KT centroid from the same dataset used in C and D. Means are represented by "+." ****, P < 0.0001; ***, P < 0.001; **, P < 0.01; *, P < 0.05; ns, nonsignificant; unpaired two-tailed t test. AU, arbitrary units. KT, kinetochore.

including in the vicinity of KTs, impacts chromosome segregation fidelity.

### Mitotic errors due to excessive α-tubulin detyrosination result from incorrect KT-MT attachments

Persistent merotelic attachments in which a single KT is attached to MTs from both spindle poles are the main source of anaphase lagging chromosomes in mammalian cells (Cimini et al., 2003). To investigate whether the lagging chromosomes observed after experimental increase in α-tubulin detyrosination result from the formation of merotelic attachments, we used super-resolution coherent hybrid stimulated emission depletion (CH-STED) microscopy (Pereira et al., 2019) in fixed cells after TTL RNAi. Relative to conventional 2D-STED, this novel subdiffraction technique improves contrast in highly complex environments, such as the mitotic spindle. Accordingly, we observed a significant increase in lagging chromosomes with single stretched KTs attached to MTs from both poles, confirming the formation of merotelic attachments (Fig. 2, G and H). Of note, TTL-depleted cells also showed a significant increase in acentric (i.e., without KTs) chromosome fragments, probably resulting from the sheer stress undergone by the centromere during correction of merotelic attachments during anaphase (Cimini et al., 2003; Guerrero et al., 2010) or due to chromosome shattering of missegregated chromosomes in the previous division cycle (Crasta et al., 2012). Taken together, these data indicate that a significant fraction of the lagging chromosomes observed after experimental increase of α-tubulin detyrosination derive from erroneous merotelic attachments.

### Excessive α-tubulin detyrosination impairs efficient error correction during mitosis

To distinguish whether increased α-tubulin detyrosination increases error formation and/or prevents efficient error correction, we induced the assembly of monopolar spindles with monastrol, which favors the formation of merotelic attachments during spindle bipolarization after monastrol washout (Cimini et al., 2003; Kapoor et al., 2000). This treatment led to an equivalent increase in lagging chromosomes in both control and TTL-depleted anaphase cells (Fig. 2 I), suggesting that the increased frequency of mitotic errors observed after experimental increase of α-tubulin detyrosination were not due to an increased error formation rate. This conclusion was substantiated by the observation that the timing of centrosome separation at nuclear envelope breakdown, an important variable that has been linked to increased error formation during mitosis (Silkworth et al., 2012), was indistinguishable between control and TTL-depleted cells (Fig. S4, A–C). Thus, increased α-tubulin detyrosination impairs efficient error correction during mitosis.

### Excessive α-tubulin detyrosination does not interfere with global kMT dynamics

Given that α-tubulin detyrosination correlates with MT stability, which has been proposed as the main cause of persistent merotelic attachments behind chromosomal instability (Bakhoum et al., 2009; Cimini et al., 2003, 2006), we investigated whether constitutive amplification of α-tubulin detyrosination globally affects kMT stability. To do so, we used fluorescence dissipation after photoactivation (FDAPA) of GFP-α-tubulin (Fig. 4 A) to measure kMT turnover in control and TTL-depleted cells, as well as in MCAK-depleted cells used as positive control (Bakhoum et al., 2009; Ferreira et al., 2018). The resulting decay curves for each individual cell fit a double exponential ($R^2$ > 0.99), reflecting the presence of two MT populations: a fast fluorescence decay population that has been attributed to non-kMTs, and a slow fluorescence decay population thought to correspond to more stable kMTs (Zhai et al., 1995). From each fit we calculated and plotted the mean ± SD of the fluorescence intensity for each time point in the different conditions (Fig. 4 B), and extracted the relative percentages of stable (kMTs) vs. unstable (non-kMTs) MTs, as well as their respective $t_{1/2}$. We found that only MCAK depletion resulted in a statistically significant decrease in the proportion of kMTs in both prometaphase and metaphase (Fig. 4 C). KMT turnover also increased from prometaphase to metaphase in all conditions, but it was indistinguishable between control and TTL-depleted cells at each given stage (Fig. 4 D). In line with these data, measurement of inter-KT distances as a proxy for tension on amphitelic attachments was also indistinguishable between control and TTL-depleted cells (Fig. S1 D). In contrast, MCAK-depleted cells had slightly more stable kMTs specifically during metaphase (Fig. 4 D), as shown previously (Bakhoum et al., 2009). Altogether, these data are consistent with previous measurements of MT $t_{1/2}$ after experimental increase of α-tubulin detyrosination in interphase (Webster et al., 1990), and demonstrate that, contrary to MCAK, α-tubulin detyrosination does not globally interfere with kMT dynamics.

### Impaired error correction due to excessive α-tubulin detyrosination results from compromised MCAK activity

Interestingly, while TTL or MCAK depletion significantly increased the frequency of anaphase cells with lagging chromosomes, TTL-depleted cells showed a clear enrichment of detyrosinated α-tubulin along kMTs, whereas MCAK-depleted cells strongly accumulated detyrosinated α-tubulin preferentially on astral MTs (Fig. 5, A–D). These striking differences in the distribution of detyrosinated α-tubulin imply that global perturbation of the two proteins impacts kMT attachments through different mechanisms. Indeed, MCAK was proposed to

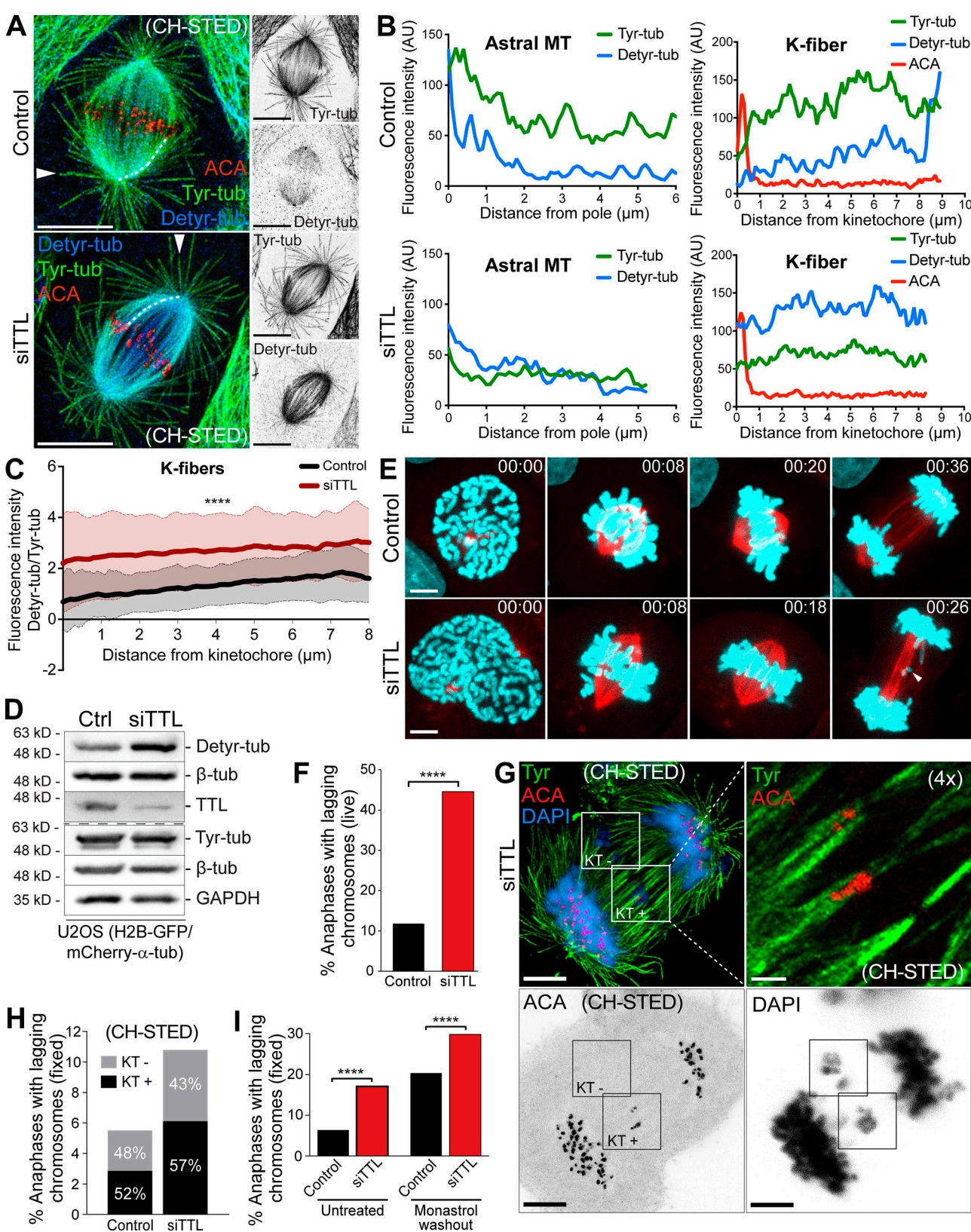

Figure 2. **Constitutively high α-tubulin detyrosination in the vicinity of KTs impairs faithful chromosome segregation. (A)** Confocal/CH-STED analysis of U2OS cells by immunofluorescence for detyrosinated tubulin (confocal), tyrosinated tubulin (CH-STED), and KTs/ACA (CH-STED). Dashed lines and arrowheads indicate representative line-scanned k-fibers and astral MTs, respectively. Scale bars, 5 μm. **(B)** Graphic representation of the line scans indicated in A. **(C)** Quantification of the fluorescence intensity ratio between detyrosinated/tyrosinated α-tubulin along k-fibers (*n* = 150 k-fibers; ~10 k-fibers/cell; five cells/experiment, three independent experiments for each condition). "0" corresponds to the KT centroid. The mean values are represented by a thicker line

and the SD associated to each point is represented by a shaded band. **(D)** Protein lysates of U2OS cells stably expressing H2B-GFP/mCherry-α-tubulin 72 h after RNAi transfection were immunoblotted for detyr-tub, tyr-tub, TTL, and β-tubulin. GAPDH was used as loading control. **(E)** Representative images from spinning-disk confocal time-series illustrating different stages of mitosis in control and TTL RNAi U2OS cells stably expressing H2B-GFP (cyan)/mCherry-α-tubulin (red). Pixels were saturated to allow the visualization of lagging chromosomes (arrowhead). Scale bars, 5 µm. Time is hours:minutes. **(F)** Quantification of the percentage of anaphase cells with lagging chromosomes in control (11.7%) and TTL-deficient (siTTL; 44.3%) U2OS H2B-GFP/mCherry-α-tubulin cells (n > 60 cells, pool of three independent experiments. ****, P < 0.0001, logistic regression). **(G)** Confocal/CH-STED analysis of a representative anaphase U2OS cell after siTTL by immunofluorescence for tyrosinated tubulin (CH-STED) and KTs/ACA (CH-STED). DNA was counterstained with DAPI (confocal). Inserts highlight KT negative (KT-) and positive (KT+) lagging chromosomes. Scale bars, 5 µm. A 4× magnified maximum projection (five z-stacks) of the insert KT+ highlights the bi-orientation and stretching of a merotelic KT. Scale bars, 1 µm. **(H)** Quantification of the relative percentage of KT- and KT+ lagging chromosomes in control and TTL RNAi cells (n > 1,000 anaphase cells, pool of two independent experiments, three replicates per experiment). **(I)** Quantification of the percentage of anaphase cells with lagging chromosomes in control and TTL-depleted (siTTL) cells in an asynchronous (untreated) cell population (control: n = 566 anaphase cells; siTTL: n = 386 anaphase cells, pool of three independent experiments. ****, P < 0.0001, logistic regression) and upon monastrol washout (control: n = 3,977 anaphase cells; siTTL: n = 3,130 anaphase cells, pool of 11 independent experiments. ****, P < 0.0001, logistic regression). AU, arbitrary units.

control the formation of robust KT-MT attachments both by regulating the length of non-kMTs through a role at MT plus ends (Domnitz et al., 2012), and by controlling chromosome directional switching by a specific role at centromeres (Kline-Smith et al., 2004; Wordeman et al., 2007). Additionally, α-tubulin detyrosination was recently shown to regulate MCAK activity required for normal astral MT length (Liao et al., 2019).

To address whether and how MT detyrosination impairs MCAK activity required for mitotic error correction, we set up rescue experiments either by overexpressing exogenous full-length EGFP-MCAK (MCAKFL) or by enhancing endogenous MCAK activity with the small-molecule agonist UMK57 (Orr et al., 2016) in TTL-deficient cells. We found that both treatments partially rescued the frequency of anaphase cells with lagging chromosomes observed after TTL depletion (Fig. 6, A–D). Similar findings were obtained after overexpression of exogenous Kif2b, another kinesin-13 with MT depolymerizing activity that localizes to prometaphase KTs (Manning et al., 2007), but not overexpression of exogenous Kif2a, a third kinesin-13 associated with MT minus ends at spindle poles (Ganem and Compton, 2004; Fig. S5, A and B), as predicted by models in which global destabilization of KT-MT attachments is able to rescue mitotic errors (Bakhoum et al., 2009). However, because our measurements of kMT $t_{1/2}$ after TTL depletion indicated that the mitotic errors caused by increased α-tubulin detyrosination do not result from an overall increase in kMT stability, our results are better explained by an alternative, nonmutually exclusive model, in which α-tubulin detyrosination compromises the activity of a specific pool of MCAK, without globally changing kMT dynamics.

### Centromeric MCAK activity is required and sufficient to correct mitotic errors caused by a constitutive increase of α-tubulin detyrosination, without globally changing kMT dynamics

To distinguish between both models, we performed additional rescue experiments with MCAK mutant constructs that compromise MT depolymerizing activity (MCAKhypir) or centromeric localization (ΔNMCAK; Domnitz et al., 2012; Wordeman et al., 2007). These experiments suggested that α-tubulin detyrosination impairs error correction by specifically inhibiting MCAK activity at centromeres (Fig. 6, A–C and E). To further test

this hypothesis, we constitutively targeted a GFP-tagged minimal motor domain of MCAK to the centromere fused with the centromere binding domain of CENP-B, with (GCPBM) or without (GCPBMhypir) MT depolymerizing activity (Wordeman et al., 2007). We found that overexpression of either GCPBM or GCPBMhypir did not cause any measurable differences in mitotic spindle and astral MT length (Fig. 6, A–C, F, and G). Yet GCPBM, but not its inactive counterpart, was sufficient to significantly rescue the abnormally high frequency of anaphase cells with lagging chromosomes observed in TTL-depleted cells (Fig. 6 E), suggesting that, in addition to affecting MCAK's role on astral MTs (Liao et al., 2019), α-tubulin detyrosination also inhibits a restricted pool of centromeric MCAK required for error correction. To directly test this, we used FDAPA in U2OS cells stably expressing photoconvertible mEos-α-tubulin to measure global kMT $t_{1/2}$ after overexpression of GCPBM. We found that overexpression of GCPBM abolished the stabilization of kMTs typically observed from prometaphase to metaphase, without significantly affecting overall kMT $t_{1/2}$ (Fig. 7, A–C). Thus, MCAK depolymerizing activity at centromeres is required and sufficient to correct mitotic errors caused by a constitutive increase of α-tubulin detyrosination, independently from its global role in the regulation of spindle MT dynamics.

## Discussion

The prevalent view is that MTs are "passive" elements during mitotic error correction. The results reported in this work support a model in which differential α-tubulin tyrosination/detyrosination on kMTs allows MT depolymerizing enzymes localized at centromeres/KTs to discriminate between correct and incorrect kMT attachments. According to this model (Fig. 7 D), initial attachments (including incorrect ones) established during early mitosis are essentially formed by tyrosinated MTs, and thus are more permissive to depolymerize and detach. As chromosomes bi-orient and centromeric tension develops, amphitelic attachments become more stabilized and consequently more detyrosinated, leading to their further stabilization. This implies that the kinetics of α-tubulin detyrosination is sufficiently fast to accumulate near the MT plus ends as they become stably attached at KTs. Indeed, TTL was shown to only retyrosinate free tubulin dimers (Arce et al., 1975; Raybin and

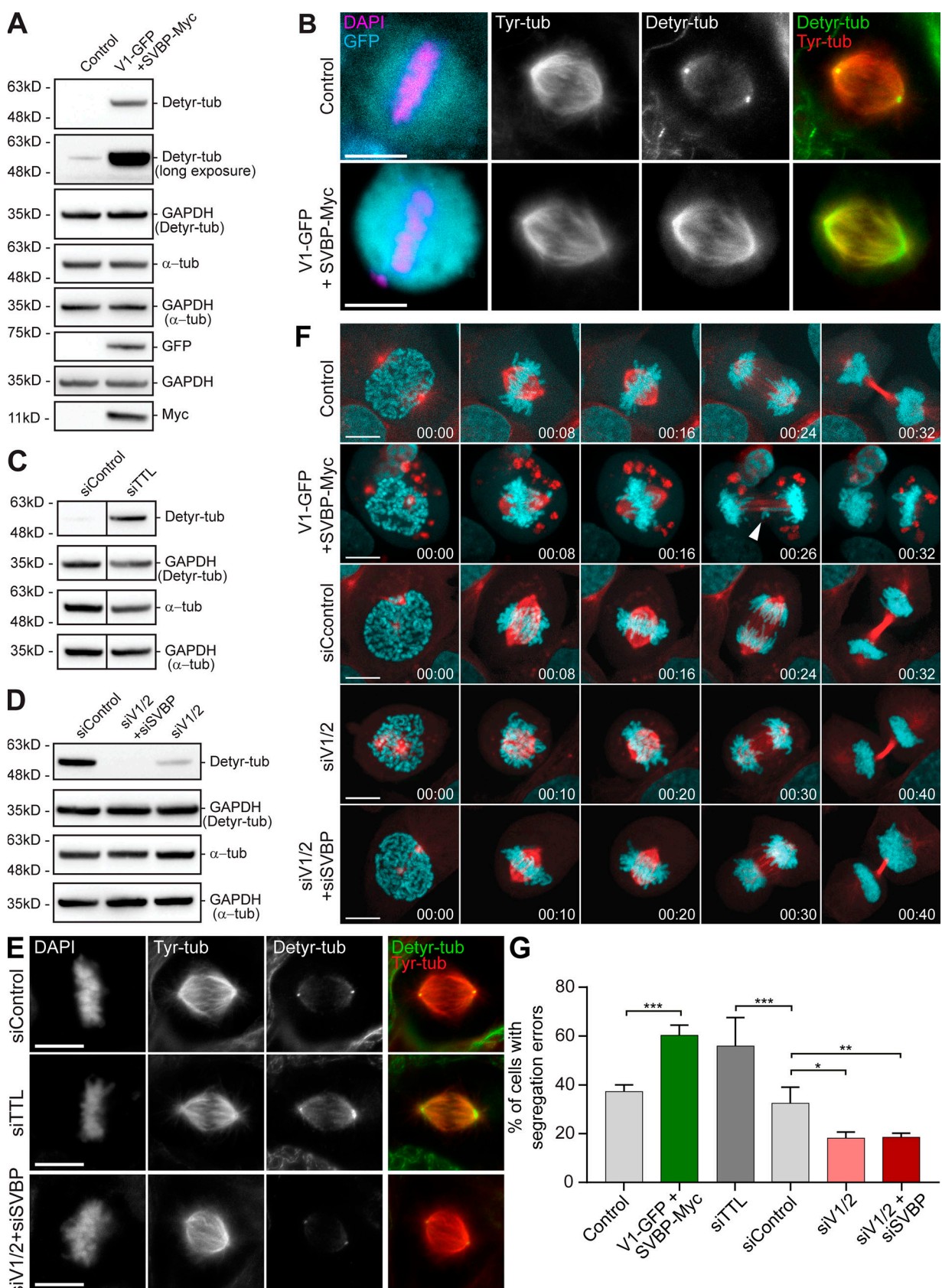

Figure 3. **Modulation of α-tubulin detyrosination levels by manipulation of vasohibins- SVBP impacts chromosome segregation fidelity. (A, C, and D)** Immunoblot analysis of detyrosinated α-tubulin and total α-tubulin levels after VASH1-SVBP overexpression, RNAi mediated knockdown of TTL or VASH1 and VASH2 (with/without SVBP), respectively. GAPDH was used as loading control. The expression of VASH1-GFP and SVBP-myc was determined using

anti-GFP and anti-myc antibodies, respectively. **(B and E)** Representative scanning confocal microscopic images of metaphase spindles immunostained with antibodies against detyrosinated and tyrosinated α-tubulin for each condition. DNA was counterstained with DAPI. Scale bars, 10 µm. **(F)** Representative images from spinning-disk confocal time-series displaying different stages of mitosis in U2OS cells stably expressing H2B-GFP (cyan)/mCherry-α-tubulin (red) transfected with the indicated plasmids and siRNAs. White arrowhead points to a segregation error during anaphase. Scale bars, 10 µm. Time in hours:minutes. **(G)** Quantification of the percentage of anaphase cells with segregation defects (lagging chromosomes and DNA bridges not discriminated) from spinning-disk confocal imaging of U2OS cells stably expressing H2B-GFP/mCherry-α-tubulin transfected with the indicated plasmids and siRNAs. Error bars indicate SD of the mean. N (number of cells, number of independent experiments): control(GFP)(38, 4); VASH1-GFP+SVBP-myc (38, 4); siTTL (28, 2); siControl (121, 5); siVASH1/2 (91, 4); and siVASH1/2+siSVBP (45, 3). *, P < 0.05; **, P < 0.01; ***, P < 0.001; one-way ANOVA.

Flavin, 1975), and recent in vitro essays revealed that α-tubulin detyrosination by VASH1/VASH2-SVBP on polymerized MTs occurs within seconds and reaches steady-state in ~10 min (Aillaud et al., 2017; van der Laan et al., 2019). Importantly, VASH1/SVBP also showed substantial detyrosinating activity toward the free tubulin dimer, whereas VASH2/SVBP appeared to specifically modify polymerized MTs, and it did so with a much higher overall detyrosinating activity (van der Laan et al., 2019). If a similar behavior is present in cells, it is conceivable that a fraction of α-tubulin detyrosination occurs even before

the free dimer is incorporated into polymerized MTs, increasing exponentially as correctly attached MTs become stabilized. While allowing the selective preservation of correct amphitelic attachments, the proposed MT-based positive feedback mechanism explains how de novo merotelic attachments on already bi-oriented chromosomes can be surgically prevented/corrected, given that the low α-tubulin detyrosination of newly attached MTs would favor their selective destabilization. Thus, spindle MTs are not just passive elements during mitotic error correction, and their detyrosination/tyrosination works as an active

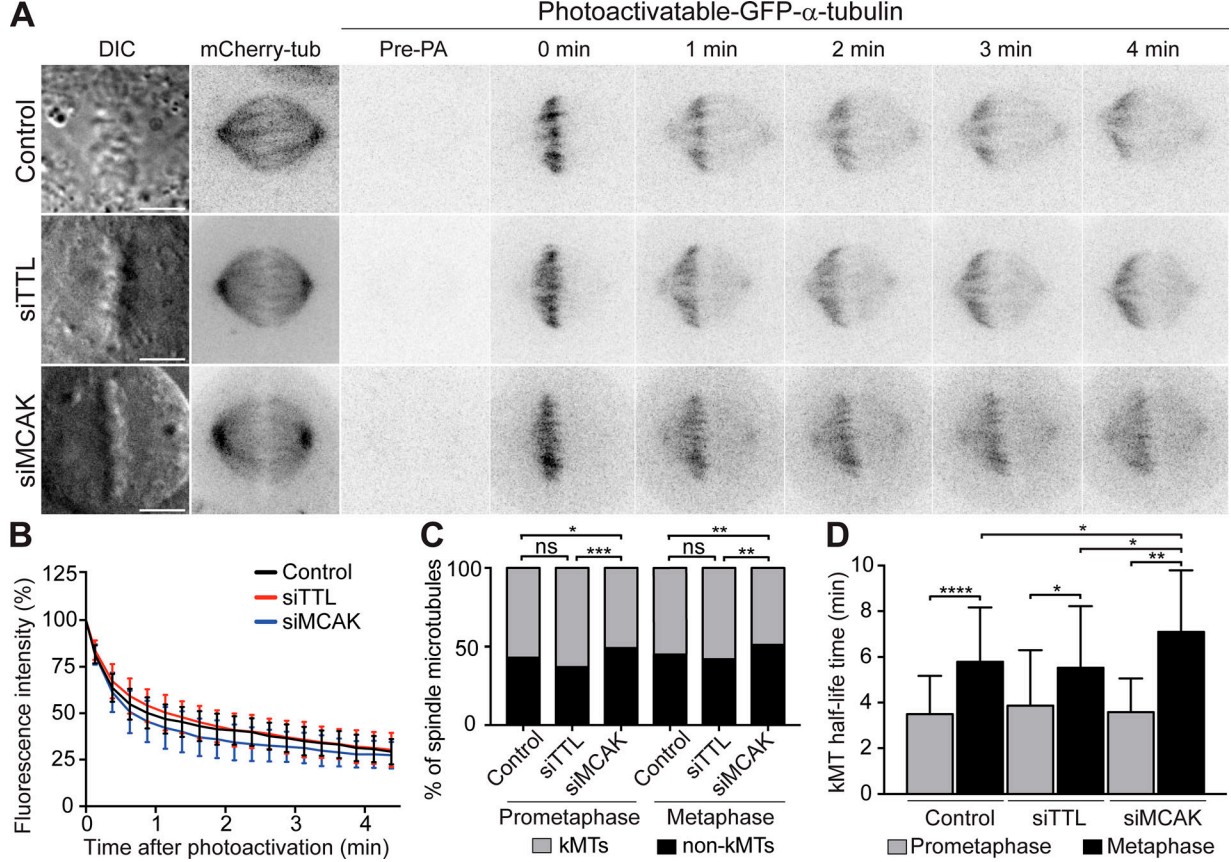

Figure 4. **MT detyrosination does not interfere with global kMT dynamics. (A)** Representative images of DIC and time-lapse fluorescent images of metaphase spindles in control, TTL-deficient (siTTL) and MCAK-deficient (siMCAK) U2OS cells stably expressing photoactivatable-GFP/mCherry-tubulin before photoactivation (Pre-PA) and at the indicated time points (min) after photoactivation of GFP-tubulin fluorescence. Scale bars, 5 µm. **(B)** Normalized fluorescence intensity after photoactivation in all conditions for metaphase cells. Data points represent mean ± SD. **(C)** Relative distribution of kMTs and non-kMTs obtained for each experimental condition after photoactivation **(D)** Calculated kMT $t_{1/2}$ under different conditions. Bars indicate mean values/cell, and error bars represent SD. Control-prometaphase: n = 31, pool of five independent experiments; control-metaphase: n = 32 cells, pool of six independent experiments; siTTL-prometaphase: n = 28, pool of six independent experiments; siTTL-metaphase: n = 17 cells, pool of three independent experiments; siMCAK-prometaphase: n = 15, pool of three independent experiments; siMCAK-metaphase: n = 30 cells, pool of two independent experiments. *, P < 0.05; **, P < 0.01; ***, P < 0.001; ****, P < 0.0001; ns, nonsignificant; unpaired two-tailed t test.

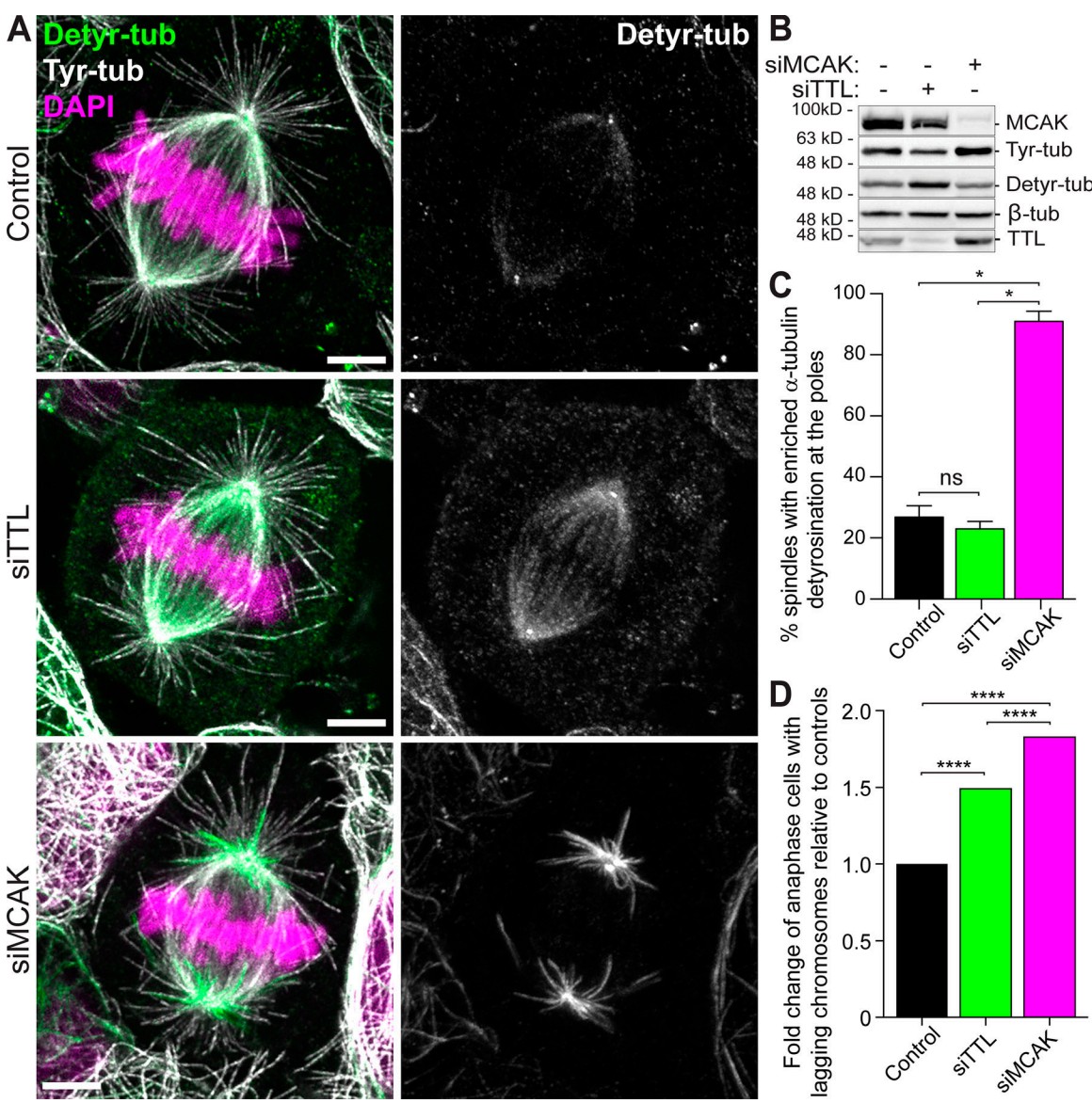

Figure 5. **Distinct distribution of detyrosinated α-tubulin after TTL or MCAK depletion. (A)** Representative scanning confocal microscopic images of metaphase spindles of U2OS cells immunostained with antibodies against detyrosinated and tyrosinated α-tubulin for each condition. DNA was counterstained with DAPI. Scale bars, 5 µm. **(B)** Protein lysates from parental U2OS cells 72 h after RNAi were immunoblotted for MCAK, detyrosinated and tyrosinated. α-tubulin, β-tubulin, and TTL. β-Tubulin were used as loading control. **(C)** Quantification of the percentage of spindles with enriched detyrosinated α-tubulin at the poles in all conditions (control: $n$ = 697 anaphase cells, siTTL: $n$ = 678 anaphase cells, siMCAK: $n$ = 608 anaphase cells, pool of three independent experiments for each condition; *, P < 0.05, logistic regression). **(D)** Quantification of the fold change of anaphase cells with lagging chromosomes after monastrol washout for each condition relative to controls (control: $n$ = 877 anaphase cells, siTTL: $n$ = 1,142 anaphase cells, siMCAK: $n$ = 743 anaphase cells, pool of three independent experiments for each condition. ****, P < 0.0001, logistic regression).

signaling mechanism that discriminates mitotic errors. Such mechanism might have evolved to promote mitotic fidelity in vertebrates, since known TTL and/or VASH-SVBP orthologues are absent from fungi, *Caenorhabditis elegans*, and *Drosophila melanogaster* (Prota et al., 2013; Sanchez-Pulido and Ponting, 2016). In these systems, mitotic fidelity might be ensured by alternative mechanisms, including the closed or semi-open nature of mitosis, or functionally distinct KTs (Drechsler and McAinsh, 2012). Last, our findings have implications for the use of MT stabilizing drugs, such as taxol and its derivatives, in the treatment of human cancers, many of which acquire

resistance by modulating MCAK activity (Ganguly et al., 2011a; Ganguly et al., 2011b).

## Materials and methods
### Cell lines and plasmids
All cell lines were cultured at 37°C in 5% $CO_2$ atmosphere in DMEM (Gibco, Thermo Fisher Scientific) containing 10% FBS (Gibco, Thermo Fisher Scientific). Parental, mEos-α-tubulin and H2B-GFP/mCherry-α-tubulin U2OS cells were kindly provided by S. Geley (Innsbruck Medical University, Innsbruck, Austria),

Figure 6. **MCAK depolymerizing activity at centromeres is required and sufficient to rescue error correction in response to a constitutive increase of MT detyrosination. (A)** Diagram of the MCAK constructs overexpressed in this study. *** indicates point mutations that inactivate the motor domain. **(B)** Protein lysates of U2OS cells 24–36 h after DNA transfection with the different MCAK constructs after TTL depletion were immunoblotted for GFP,

endogenous (endo) and tagged ectopic MCAK, Tyr-tub, Detyr-tub, and β-tubulin. GAPDH was used as loading control. **(C)** 3D-deconvolution analysis of U2OS cells overexpressing different EGFP-MCAK constructs and immunostained for tyrosinated α-tubulin (red-hot look-up table). Scale bars, 5 μm. **(D)** Quantification of the fold change of anaphase cells with lagging chromosomes relative to controls in rescue experiments using MCAKFL overexpression ($n > 1{,}500$ anaphase cells, pool of five independent experiments. ****, $P < 0.0001$, logistic regression) and MCAK pharmacological enhancement with UMK57 ($n > 1{,}000$ anaphase cells, pool of four independent experiments. ****, $P < 0.0001$, logistic regression). **(E)** Quantification of the fold change of anaphase cells with lagging chromosomes relative to controls in rescue experiments using different EGFP-MCAK constructs. Significant rescue was observed upon overexpression of MCAKFL ($n > 1{,}500$ anaphase cells, pool of five independent experiments. ****, $P < 0.0001$, logistic regression) and GCPBM ($n > 900$ anaphase cells, *, $P = 0.011$, pool of four independent experiments, logistic regression). **(F)** Quantification of the metaphase spindle length 24–48 h after transfection with different EGFP-MCAK constructs (GFP-control: $n = 47$ cells, MCAKFL: $n = 27$ cells, MCAKhypir: $n = 47$ cells, ΔNMCAK: $n = 33$ cells, GCPBM: $n = 46$ cells, GBPBMhypir: $n = 38$ cells, pool of three independent experiments for each condition, except for GCPBMhypir [pool of two independent experiments]. **, $P < 0.01$, unpaired two-tailed $t$ test). **(G)** Quantification of the astral MT length (~10 astral MTs/cell; GFP-control: $n = 471$ astral MTs, MCAKFL: $n = 264$ astral MTs, MCAKhypir: $n = 470$ astral MTs, ΔNMCAK: $n = 326$ astral MTs, GCPBM: $n = 456$ astral MTs, GBPBMhypir: $n = 378$ astral MTs, pool of three independent experiments for each condition, except for GCPBMhypir [pool of two independent experiments]. ****, $P < 0.0001$, unpaired two-tailed $t$ test). MT, microtubule.

whereas the PA-GFP-α-tubulin/mCherry-α-tubulin U2OS cells were kindly provided by R. Medema (Netherlands Cancer Institute, Amsterdam, The Netherlands). Stable H2B-GFP/mRFP-α-tubulin U2OS cells were generated in this study by lentiviral transduction.

### Constructs and transfections

To express fluorescently tagged proteins, cells were transfected using Lipofectamine 2000 (Thermo Fisher Scientific) with 0.25–2 μg of each construct. MCAK constructs (kindly provided by L. Wordeman, University of Washington, Seattle, WA): EGFP-MCAK full-length, EGFP-MCAKhypir, EGFP-ΔN-MCAK, GCP BM, and GCPBMhypir (Domnitz et al., 2012; Wordeman et al., 2007); human VASH1 (NM_014909.5), and SVBP (NM_199342.4) encoding gateway-compatible vectors (kind gift from J. Nilsson, Center for Protein Research, Copenhagen, Denmark) were used to create expression plasmids with C-terminal EGFP and 3X-myc tags by recombination between attL and attR sites (Invitrogen) according to the manufacturer's instructions. Plasmid-encoding EGFP alone was used as control. Expression of VASH1 and SVBP was performed as described previously (Liao et al., 2019). Briefly, cells were transfected with 1 μg of the two plasmids using GeneJuice (Merck) transfection reagent according to the manufacturer's protocol. To perform RNAi-mediated protein depletion, cells were plated at 30–50% confluence onto 22 × 22-mm no. 1.5 glass coverslips and cultured for 1–2 h in DMEM supplemented with 5% of FBS before transfection. RNAi transfection was performed using Lipofectamine RNAiMAX reagent (Thermo Fisher Scientific) with 50 nM of validated siRNA oligonucleotides against human TTL 5′-GUGCACGUGAUCCAGAAA U-3′, MCAK 5′-GAUCCAACGCAGUAAUGGU-3′, VASH1 5′-CCA GACUAGGAUGCUUCUG-3′, VASH2 5′-GGAUAACCGUAACUG AAGU-3′, SVBP 5′-GCAGAGAUCUAUGCCCUCA-3′, and control (scramble) siRNA 5′-UGGUUUACAUGUCGACUAA-3′, diluted in serum-free medium (Opti-MEM, Thermo Fisher Scientific). Mock transfected or scramble siRNA results were indistinguishable. Validation of TTL depletion by RNAi was obtained by expressing exogenous human TTL-YFP (kindly provided by C. Janke, Institut Curie, Orsay, France) resistant to siRNA depletions. An siRNA-resistant TTL-YFP (TTL-YFP$^R$) was obtained by site-directed mutagenesis of five nucleotides (indicated with asterisks) at the target site using the following complementary primers: siTTLresist.Fw 5′-CAACCAGGGCCAAGTC*CAT*GTC* ATT*CAA*AAATATCTTGAGCACCCTC-3′ and siTTLresist.Rev

5′-GAGGGTGCTCAAGATATTTT*TGA*ATG*ACA*TGG*ACTTGG CCCTGGTTG-3′. The same protocol was used to introduce siRNA resistance in a plasmid encoding for the catalytic dead version (E331Q; Prota et al., 2013; Szyk et al., 2011) of human TTL-YFP (TTL[CD]-YFP$^R$), kindly provided by C. Janke. The TTL knockout cells were generated by CRISPR/Cas9-mediated genome editing, using a lentiviral backbone containing both the *Streptococcus pyogenes* Cas9 (*sp*Cas9) nuclease and the single guide RNA scaffold (lentiCRISPRv2). Two 20-bp single guide RNA (sgTTL, 5′-CACCGAACAGCAGCGTCTACGCCG-3′ and sgCtrl, 5′-AAACCG GCGTAGACGCTGCTGTTC-3′; Sigma-Aldrich) were designed to target the TTL gene and cloned into the lentiviral vector pLenti-CRISPR-v2 (no. 52961, addgene). All lentivirus was produced by cotransfection of HEK293T cells with the lentiviral vectors, psPAX2 (Gag, Pol, Rev, and Tat expressing packaging vector) and pMD2.G (VSV-G expressing envelope vector). The 72 h supernatant containing DNA particles was used to transduce the lentivirus to the target cells, as previously described (Ferreira et al., 2018). TTL knockout cells were selected by their resistance to puromycin (2 μg/ml, Calbiochem) and confirmed by Western blot analysis.

### Drug treatments

To induce lagging chromosomes by inhibition of kinesin-5, we added 100 μM Monastrol (Tocris, Bioscience) 12–16 h before washout. MT stabilization was obtained using 10 nM Taxol for 1 h (Sigma-Aldrich). Mitotic arrest at metaphase was obtained using 10 μM MG132 (Calbiochem). Live-cell analysis using MG132 was performed in less than 2 h in the presence of the drug to avoid cohesion fatigue. MCAK activity was enhanced using 250 μM of UMK57 (Orr et al., 2016) for 12–16 h before fixation. An equivalent volume of DMSO was used as control for each drug treatment.

### Immunofluorescence

U2OS cells were fixed with cold methanol (–20°C) for 4 min or 0.1–0.25% gluteraldehyde + 4% paraformaldehyde (Electron Microscopy Sciences) for 10 min. Autofluorescence was quenched by 0.1% sodium borohydride (Sigma-Aldrich) after aldehyde fixation and cells permeabilized with 0.5% Triton X-100 (Sigma-Aldrich) for another 10 min. Tyrosinated α-tubulin and detyrosinated α-tubulin were immunostained using rat monoclonal anti-tyrosinated α-tubulin clone YL1/2 (1:100–500, Bio-Rad) and rabbit polyclonal anti-detyrosinated α-tubulin (1:250–2,000; Liao et al., 2019), respectively. Other primary antibodies used were

**Figure 7. Centromeric MCAK does not significantly alter global kMT dynamics. (A)** Representative images of DIC and time-lapse fluorescent images of metaphase spindles of U2OS cells stably expressing mEos-α-tubulin. Photoconversion was performed after overexpression of GFP (control) and GCPBM. Images before photoconvertion (Pre-PC) and at the indicated times points (min) after photoconversion are displayed. Scale bars, 5 µm. **(B)** Normalized fluorescence intensity after photoconversion of mEos-α-tubulin in U2OS cells in metaphase (GFP-control: $n$ = 9 cells from five independent experiments and GCPBM: $n$ = 6 cells from three independent experiments). Data points represent mean ± SD. **(C)** Calculated kMT $t_{1/2}$ in control and GCPBM-overexpressing cells in prometaphase (GFP-control: $n$ = 21 cells, and GCPBM: $n$ = 25 cells, pool of five independent experiments, P = 0.0847, unpaired two-tailed $t$ test) and metaphase (GFP-control: $n$ = 9 cells from five independent experiments and GCPBM: $n$ = 6 cells from three independent experiments, P = 0.4351, unpaired two-tailed $t$ test) using the fluorescence decay after photoconversion. Only kMT $t_{1/2}$ between prometaphase and metaphase in control cells varied significantly. ***, P < 0.001. Bars indicate mean values/cell, and error bars represent SD. **(D)** Proposed model for the discrimination of mitotic errors by MT detyrosination (see Discussion for description). O.E., overexpression.

human anti-centromere antibodies (ACA, 1:100–2,000, kind gift from B. Earnshaw, Welcome Trust Centre for Cell Biology, University of Edinburgh, Edinburgh, UK, or S. Geley). GFP-tagged constructs were visualized by means of direct fluorescence. Alexa Fluor 488 (1:200–2,000, Themofisher), STAR-580, and STAR-RED (1:200, Abberior) were used as secondary antibodies, and DNA was counterstained with 1 µg/ml DAPI (Sigma-Aldrich).

**Image acquisition**

3D wide-field images were acquired using an AxioImager Z1 (100× Plan-Apochromatic oil differential interference contrast

objective lens, 1.46 NA, Carl Zeiss Microimaging Inc.) equipped with a CCD camera (ORCA-R2, Hamamatsu) operated by Zen software (Carl Zeiss, Inc.). Blind deconvolution of 3D image datasets was performed using Autoquant X software (Media Cybernetics). Confocal and super-resolution CH-STED images were acquired using Abberior Instruments "Expert Line" gated-STED coupled to a Nikon Ti microscope. An oil-immersion 60× 1.4 NA Plan-Apo objective (Nikon, Lambda Series) and pinhole size of 0.8 Airy units were used in all confocal acquisitions. Super-resolution images were acquired using a CH-STED beam (Pereira et al., 2019). A second LSM800 confocal microscope

(Carl Zeiss Microimaging Inc.) mounted on a Zeiss-Axio imager Z1 equipped with plan-apochromat 63×/1.40 oil differential interference contrast (DIC) M27 objective (Carl Zeiss, Inc.) was used and operated by Zen 2008 software (Carl Zeiss, Inc.).

### Quantification of the fluorescence intensity along astral MTs and KT fibers

Detyrosination/tyrosination profile along astral MTs and KT fibers (including in a sub-region 1 µm distant from the KT) was performed using ImageJ by drawing a line segment along the region of interest (ROI) followed by plotting the fluorescence intensity values (command plot profile at each channel) as a function of distance from the KT centroid determined by ACA signal. Background was quantified in a region outside the spindle and subtracted from each channel.

### Quantification of detyrosination at the KTs

The fluorescence intensity signal of tubulin detyrosination was measured directly at KTs in a 9 × 9-pixel-square ROI using LAPSO (MATLAB) and normalized to the intensity signal of ACA in the same ROI. Background fluorescence was measured outside the ROI and subtracted from each KT. The mean values of all KTs quantified in one cell were plotted in a scattered dot plot.

### Quantification of spindle and astral MT length

The spindle length was calculated as the distance between the two centrosome position adjusted to the focal plane according to the equation: spindle length = $\sqrt{[d(centrosome1 - centrosome2, xy)^2 + d(centrosome1 - centrosome2, z)^2]}$, d = distance. Astral MT length was measured as the distance between the centrosome and the MT distal tip using maximum-projection images.

### Quantification of the inter-KT distance

Inter-KT distances were measured using a custom program written in MATLAB 8.1 (QUANTA), which determines the 3D distance between the centroid of the sister KTs.

### Time-lapse microscopy

For phenotypic analysis of TTL depletion (siTTL), U2OS H2B-GFP/mCherry-α-tubulin cells were cultured in glass coverslips and assembled into 35-mm magnetic chambers (14 mm, no. 1.5, MatTek Corporation). Cell culture medium was replaced with phenol-red-free DMEM $CO_2$-independent medium (Invitrogen) supplemented with 10% FBS. Time-lapse imaging was performed in a heated chamber (37°C) using a 100× 1.4 NA Plan-Apochromatic differential interference contrast objective mounted on an inverted microscope (TE2000U; Nikon) equipped with a CSU-X1 spinning-disk confocal head (Yokogawa Corporation of America) and with two laser lines (488 nm and 561 nm). Images were acquired with an iXon+ EM-CCD camera (Andor Technology). Eleven 1-µm-separated z-planes covering the entire volume of the mitotic spindle were collected every 2 min. The phenotypic analysis of U2OS H2B-GFP/mCherry-α-tubulin cells overexpressing GFP and VASH1-GFP+SVBP-Myc, as well as the analysis of VASH1/2, VASH1/2+SVBP, and additional TTL depletion (used as positive control) in the same experimental dataset, were performed using a Plan-Apochromat

DIC 63×/1.4 NA oil objective mounted on an inverted Zeiss Axio Observer Z1 microscope (Marianas Imaging Workstation from Intelligent Imaging and Innovations Inc. [3i]), equipped with a CSU-X1 spinning-disk confocal head (Yokogawa Corporation of America), iXon Ultra 888 EM-CCD camera (Andor Technology), and four laser lines (405 nm, 488 nm, 561 nm, and 640 nm). Fifteen 1-µm-thick z-planes were collected every 2 min for 2–3 h. Image processing was performed in ImageJ. All displayed images represent maximum-intensity projections of z-stacks. Additional phenotypic live-imaging analysis of TTL KO H2B-GFP/mCherry-α-tubulin cells and respective controls was performed in an IN Cell Analyzer 2000 microscope (GE Healthcare), equipped with temperature and $CO_2$ controller, using a Nikon 40×/0.95 NA Plan Fluor objective and a large chip CCD Camera (CoolSNAP K4) with a pixel array of 2,048 × 2,048. Single planes were acquired every 2 min.

### Error correction assay

All U2OS cell lines used in this assay were seeded onto 22 × 22-mm no. 1.5 glass converslips and cultured at 37°C in 5% $CO_2$ atmosphere in DMEM (Gibco, Thermo Fisher Scientific) supplemented with 10% FBS (Gibco, Thermo Fisher Scientific) for 72 h. 100 µM of Monastrol (Tocris, Bioscience) was added 12–16 h before release, which was performed by three consecutive washes with warm PBS followed by incubation with DMEM + 10% FBS for 40–50 min. Cells were fixed and immunostained for tyrosinated α-tubulin. DNA was counterstained with DAPI. Only chromosomes spatially separated from the main mass of chromosomes were accounted as laggards.

### High-throughput screening of lagging chromosomes

Quantification of lagging chromosomes was performed using 50–500 images of contiguous fields acquired in an IN Cell Analyzer 2000 microscope (GE Healthcare) with a Nikon 40×/0.95 NA Plan Fluor objective (binning 2 × 2), using a large chip CCD Camera (CoolSNAP K4) with a pixel array of 1,024 × 1,024 (2.7027 pixel/µm resolution). All early to late anaphase figures were classified regarding the presence or absence of lagging chromosomes. Any DAPI-positive material between the two chromosome masses, but distinguishably separated from them, was counted as lagging chromosomes. DNA bridges were excluded from this analysis.

### Photoactivation (PA) and photoconversion (PC)

For PA and PC assays, both U2OS cells stably expressing PA-GFP-α-tubulin/mCherry-α-tubulin or mEos-α-tubulin, respectively, were cultured on glass coverslips as described. Mitotic cells were identified by Differential Interference Contrast (DIC) microscopy, and imaging was performed using a Plan-Apo 100× NA 1.40 DIC objective on a Nikon TE2000U inverted microscope equipped with a Yokogawa CSU-X1 spinning-disc confocal head containing two laser lines (488 nm and 561 nm) and a Mosaic (Andor) photoactivation system (405 nm). Photoactivation and photoconversion were performed in late prometaphase or metaphase cells, the latter treated with 10 µM MG132 for less than 2 h, identified by mCherry-α-tubulin or mEos-α-tubulin signal, respectively. A region of interest for PA/PC was selected using a

line segment placed perpendicular to the main axis in one of the sides adjacent to the metaphase plate. Seven 1-µm separated z-planes centered in the middle of the mitotic spindle were captured every 10–15 s for 4.5 min (a prePA/PC image was always acquired) using 561-nm and 488-nm lasers and an iXon+ EM-CCD camera. MT $t_{1/2}$ was calculated using a MATLAB algorithm for kymograph generation and analysis. Briefly, generated kymographs were collapsed, and the fluorescence intensity curve was created. Fluorescence intensity peaks of each time frame were defined by manually tracking regions of the highest fluorescence intensities followed by automated curve fitting and normalized intensities to the first time-point after PA/PC following background subtraction (background values obtained from quantifying the respective nonactivated half-spindle). Values were corrected for photobleaching by normalizing to the values obtained from the quantification of fluorescence loss of whole-cell sum projected images. MT turnover was calculated based on a fitted curve of the normalized intensities at each time point (corrected for photobleaching) to a double exponential curve $A1*exp(-k1*t) + A2*exp(-k2*t)$ using MATLAB (MathWorks), in which $t$ is time, A1 represents the less stable (non-kMTs) population, and A2 the more stable (kMTs) population with decay rates of k1 and k2, respectively (cells displaying an $R^2$ value <0.99 were excluded from quantification). From the curve, the rate constants and the percentage of MTs were extracted for the fast (typically interpreted as the fraction corresponding to non-kMTs) and the slow (typically interpreted as the fraction corresponding to kMTs) processes. The $t_{1/2}$ was calculated as ln2/k for each MT population.

### Tracking of centrosome separation

Detailed quantitative analysis of centrosome location and nuclear envelope topology was performed using custom made MATLAB scripts MathWorks, R2018a). Tracking of centrosome position/trajectories was performed in 3D space using image stacks with a pixel size of 0.190 µm and z-step of 0.7 µm. Nuclear shape was reconstructed in 3D using H2B-GFP as a marker. From the centrosome locations and nuclear envelope reconstruction, it was possible to calculate the angle between the centrosome–centrosome axis and the nucleus major axis at nuclear envelope breakdown (NEB).

### Micropatterning

10-µm-width line micropatterns to control individual cell shape and adhesion pattern were produced as follows: glass coverslips (22 × 22 mm, no. 1.5, Van Waters and Rogers) were activated with plasma (Zepto Plasma System, Diener Electronic) for 1 min and incubated with 0.1 mg/ml of PLL(20)-g[3,5]-PEG(2) (SuSoS) in 10 mM Hepes at pH 7.4 for 1 h at room temperature. After rinsing and air-drying, the coverslips were placed on a synthetic quartz photomask (Delta Mask), previously activated with deep-UV light (PSD-UV, Novascan Technologies) for 5 min. 3 µl of MiliQ water was used to seal each coverslip to the mask. The coverslips were then irradiated through the photomask with the UV lamp for 5 min. Afterward, coverslips were incubated with 25 µg/ml of fibronectin (Sigma-Aldrich) and 5 µg/ml of Alexa 647–conjugated fibrinogen (Thermo Fisher Scientific) in

100 mM NaHCO$_3$ at pH 8.6 for 1 h at room temperature. Cells were seeded at a density of 50,000 cells/coverslip and allowed to spread for ~10–15 h before imaging. Nonattached cells were removed by changing the medium ~2–5 h after seeding.

### Western blotting analysis

Cells were grown until 90% confluence in complete growth medium and harvested by centrifugation at 1,200 rpm for 5 min. Cell pellets were washed once with warm PBS and resuspended in ice-cold lysis buffer (50 mM Tris HCl, pH 7.4, 150 mM NaCl, 1 mM EDTA, 1 mM EGTA, 0.5% NP-40, and 0.5% Triton X-100) supplemented with a cocktail of protease inhibitors (Roche). Protein samples were denatured in Laemmli buffer at 95°C for 5 min, and 50 µg of total protein were separated by 10% (vol/vol) SDS-PAGE electrophoresis. Proteins were transferred to a nitrocellulose membrane using an IBlot Dry Blotting System (Invitrogen). TTL and tubulins were probed using the following antibodies: rabbit polyclonal anti-TTL (1:2,000, ProteinTech), rat monoclonal anti-tyrosinated α-tubulin clone YL1/2 (1:2,000, Bio-Rad), rabbit polyclonal anti-detyrosinated α-tubulin (Liao et al., 2019), mouse monoclonal anti-polyglutamilated tubulin (1:1,000, Adipogen), mouse monoclonal anti-α-tubulin B-5-1-2 clone (1:10,000; Sigma-Aldrich), and mouse monoclonal anti-β-tubulin clone T5201 (1:2,000, Sigma-Aldrich). Other proteins were immunodetected using mouse monoclonal anti-Cas9 (1:2,500, Merck), anti-mCherry (1:2,500, kind gift from I. Cheeseman, Massachusetts Institute of Technology, Cambridge, MA), rabbit or goat polyclonal anti-GFP (1:2,500, made by our in-house facility and 1:1,000, Rockland Immunochemicals Inc., respectively), rabbit polyclonal anti-MCAK (1:5,000, kind gift from D. Compton, Geisel School of Medicine, Dartmouth College, Hanover, NH), mouse monoclonal anti-Myc (1:10,000; Cell Signaling Technology), and mouse monoclonal anti-GAPDH (1:40,000; Proteintech). HRP-conjugated secondary antibodies (1:5,000–10,000, Jackson Immunoresearch) were visualized using an ECL system (Bio-Rad). Protein levels were quantified by chemiluminescence using a ChemicDoc XRS+ system (Bio-Rad).

### Statistical analysis

Logistic regression was performed to evaluate the impact of multiple treatments and the occurrence of lagging chromosomes during anaphase, adjusting the model for the different experiments. One of the advantages of logistic regression is that it takes into account the effect of a risk predictor (coefficient size) that evaluates the enhancement (positive coefficient) or reversal (negative coefficient) of a particular variable/phenotype. Data variation was also taken into account when the logistic regression model was adjusted for this variable. Significance was set at $\alpha = 0.05$, and these analyses were performed using SPSS, version 24.0. An unpaired two-tailed Student's $t$ test was used to determine the significance of differences between two groups (fluorescence intensity, spindle length, inter-KT distance, and MT $t_{1/2}$). The Student's $t$ test was adjusted to the significance of the variance between experiments. Comparison between multiple conditions was performed using one-way ANOVA. The non-parametric Kolmogorov–Smirnov test was used to determine the significance of differences in centrosome separation

between conditions. Significance was set at α = 0.05, and these analyses were performed using SPSS, version 24.0, and Prism, version 7.0a.

### Online supplemental material

Fig. S1 shows that TTL depletion increases α-tubulin detyrosination on astral MTs and in the vicinity of KTs without affecting inter-KT distance. Fig. S2 shows evidence for cellular adaptation to the chronic loss of TTL using CRISPR-Cas9 KO cells. Fig. S3 shows that faithful chromosome segregation requires α-tubulin retyrosination and relies on the catalytic activity of TTL. Fig. S4 shows that chromosome missegregation observed after TTL depletion is not due to altered centrosome separation at nuclear envelope breakdown. Fig. S5 shows that overexpression of a second kinesin-13 (Kif2b) also rescues chromosome missegregation due to increased α-tubulin detyrosination.

## Acknowledgments

We would like to thank Linda Wordeman for kindly sharing MCAK constructs, Ben Kwok (University of Montreal, Montreal, Quebec, Canada) and Duane Compton (Geisel School of Medicine, Dartmouth, NH) for the UMK57, Stephan Geley and René Medema for the U2OS lines, Jakob Nilsson (Novo Nordisk Foundation Center, University of Copenhagen, Copenhagen, Denmark) for gateway plasmids, Carsen Janke for the TTL-YFP plasmids, Martina Barisic (Danish Cancer Society Research Center, Copenhagen, Denmark) for technical support, and the Biosciences Screening Scientific platform at i3S.

This work was funded by the European Research Council under the European Union's Horizon 2020 research and innovation program (grant agreement no. 681443) and FLAD Life Science 2020 (Proj. 261/2014 to H. Maiato), by grants from the Danish Cancer Society Scientific Committee (KBVU; R146-A9322) and the Lundbeck Foundation (R215-2015-4081 to M. Barisic), and grants from FEDER - Fundo Europeu de Desenvolvimento Regional funds through the COMPETE 2020 - Operacional Programme for Competitiveness and Internationalization (POCI) Portugal 2020 and by Portuguese funds through Fundação para a Ciência e a Tecnologia/Ministério da Ciência, Tecnologia e Ensino Superior in the framework of the project PTDC/BEX-BCM/1758/2014 (POCI-01-0145-FEDER-016589) to J.G. Ferreira and PPBI-POCI-01-0145-FEDER-022122 to the Biosciences Screening Scientific platform at i3S. L.T. Ferreira was supported in part by a studentship from Fundação para a Ciência e a Tecnologia (SFRH/BD/79174/2011).

The authors declare no competing financial interests.

Author contributions: Investigation, formal analysis, and validation (L.T. Ferreira, B. Orr, G. Rajendraprasad, C.G. Boldú, J.T. Lima); conceptualization, supervision, and funding acquisition (H. Maiato, M. Barisic, J.G. Ferreira); resources, methodology, and software (A.J. Pereira); validation and formal analysis (C. Lemos); writing - original draft (L.T. Ferreira, H. Maiato); writing – review and editing (L.T. Ferreira, G. Rajendraprasad, C.G. Boldú, J.T. Lima, H. Maiato, M. Barisic); project administration (H. Maiato).

Submitted: 11 October 2019

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

# Supplemental material

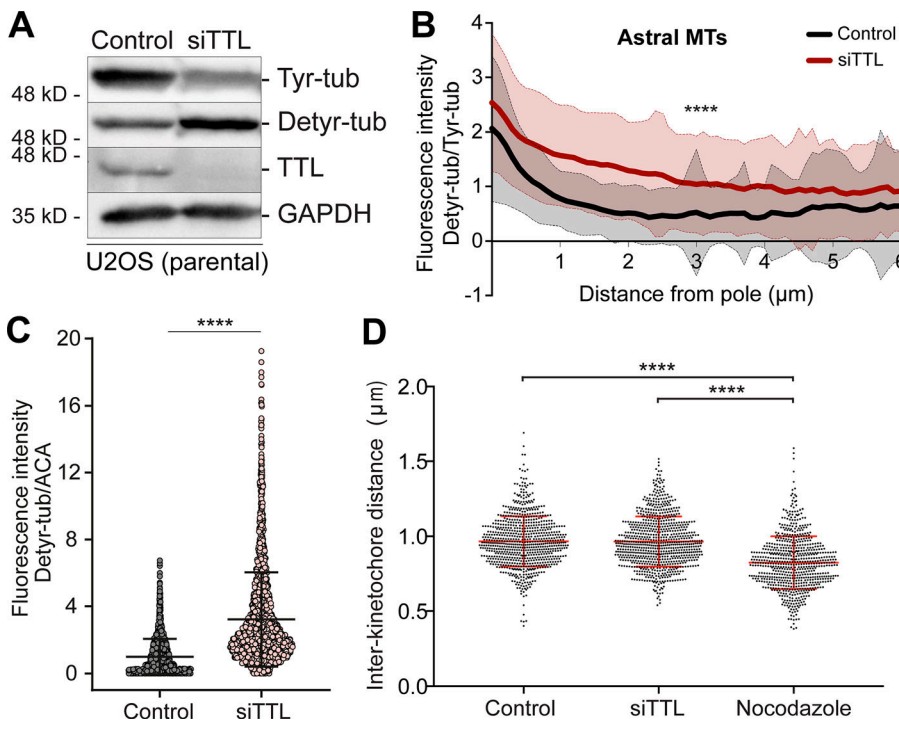

Figure S1.    **TTL depletion increases α-tubulin detyrosination on astral MTs and in the vicinity of KTs without affecting inter-KT distance. (A)** Protein lysates from parental U2OS cells collected 72 h after RNAi transfection in control and after siTTL were immunoblotted for tyrosinated tubulin, detyrosinated tubulin, and TTL. GAPDH was used as loading control. **(B)** Quantification of the fluorescence intensity ratio between detyrosinated/tyrosinated α-tubulin along astral MTs in control and siTTL metaphase cells ($n$ = 150 astral MTs/condition; 10 astral MTs/cell, 5 cells/experiment, pool of three independent experiments. ****, P < 0.0001, unpaired two-tailed $t$ test). **(C)** Quantification of the fluorescence intensity of detyrosinated α-tubulin relative to ACA at the KTs in control siTTL U2OS cells ($n$ [control] = 2,224 KTs, 19 metaphase cells, pool of two independent experiments; $n$ [siTTL] = 2,507 KTs, 19 metaphase cells, pool of two independent experiments. ****, P < 0.0001, unpaired two-tailed $t$ test). The mean fluorescence intensity of tubulin detyrosination at KTs normalized by the mean of control cells is represented. Error bars represent SD. **(D)** Quantification of the inter-KT distance in control, siTTL, and nocodazole (1 µM)–treated U2OS parental cells ($n$ [control] = 762 KT pairs, $n$ [siTTL] = 795 KT pairs, and $n$ [nocodazole] = 695 KT pairs, from 6–11 cells/condition, pool of three independent experiments. ****, P < 0.0001, unpaired two-tailed $t$ test).

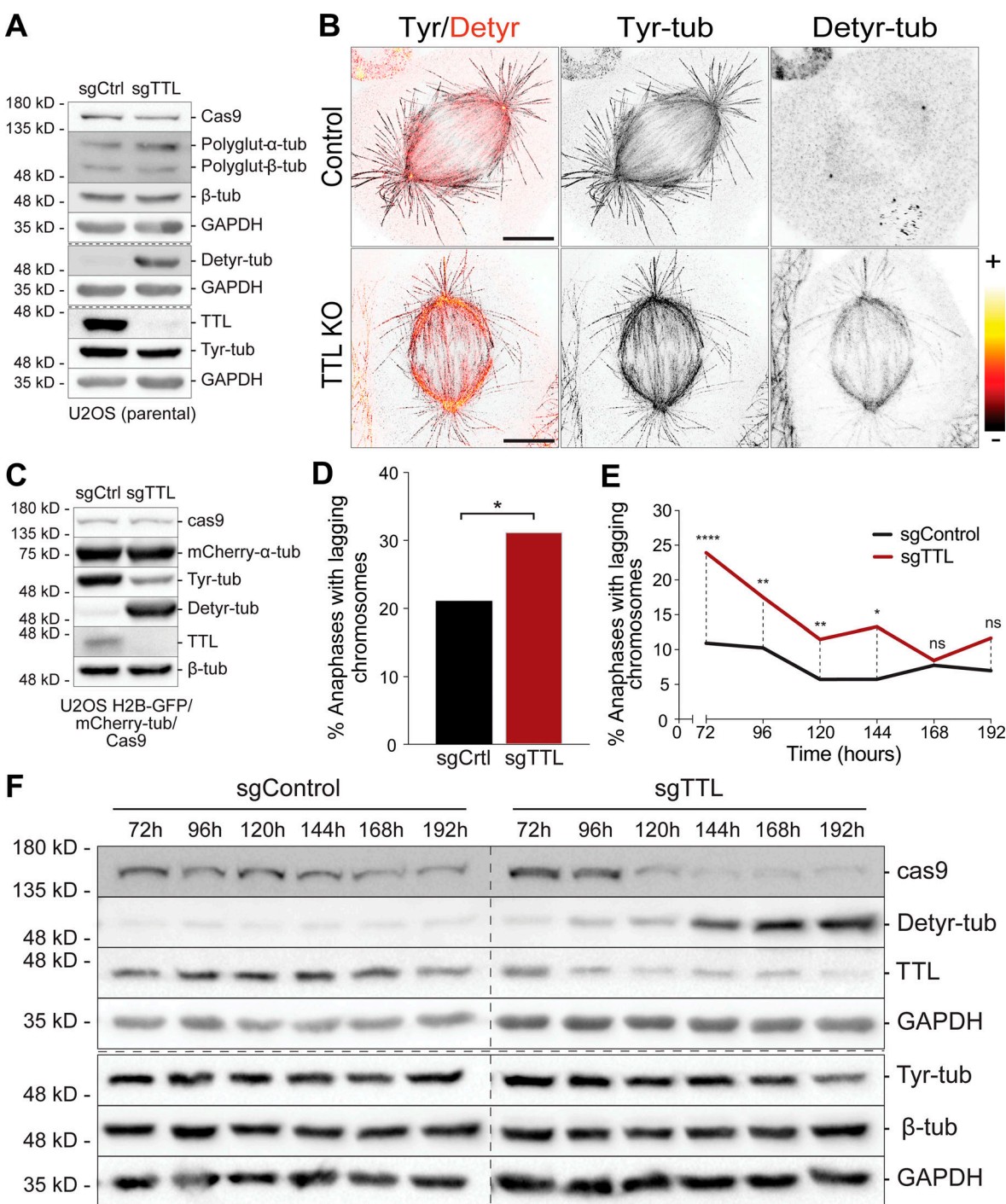

Figure S2.  **Cellular adaptation to the chronic loss of TTL. (A)** Protein lysates of control (sgCtrl) and TTL KO (sgTTL) U2OS cells were immunoblotted for cas9, polyglutamylated tubulin (polyglut-α/β-tub), α-tubulin (α-tub), TTL, tyrosinated α-tubulin (Tyr-tub), and detyrosinated α-tubulin (Detyr-tub). GAPDH was used as loading control. **(B)** Confocal/CH-STED analysis of control (sgControl) and TTL KO (sgTTL) U2OS cells by immunofluorescence for Detyr-tub (confocal) and Tyr-tub (CH-STED). Red-hot lookup table was used to map extremes in fluorescence intensity for Detyr-tub. Scale bars, 5 μm. **(C)** Protein lysates of control (sgCtrl) and TTL KO (sgTTL) U2OS H2B-GFP/mCherry-tub cells were immunoblotted for cas9, mCherry, Tyr-tub, Detyr-tub, and TTL. α-tubulin was used as loading control. **(D)** Quantification of the percentage of anaphase cells with lagging chromosomes in control (sgCtrl) and TTL KO (sgTTL) U2OS H2B-GFP/mCherry-tub cells by live-cell imaging between 72 and 168 h after lentiviral transduction (n [sgCtrl] = 266 cells, n [sgTTL] = 241 cells, pool of two independent experiments, with two and nine replicates, respectively, *, P < 0.05, logistic regression). **(E)** Quantification of the percentage of anaphase cells with lagging chromosome between 72 and 192 h after lentiviral transduction in control (sgCtrl) and TTL KO (sgTTL) U2OS cells by immunofluorescence in fixed cells (n [sgCtrl] 72 h = 324 cells, n [sgCtrl] 96 h = 311 cells, n [sgCtrl] 120 h = 280 cells, n [sgCtrl] 144 h = 264 cells, n [sgCtrl] 168 h = 320 cells, n [sgCtrl] 192 h = 356 cells; n [sgTTL] 72 h = 301 cells, n [sgTTL] 96 h = 377 cells, n [sgTTL] 120 h = 462 cells, n [sgTTL] 144 h = 256 cells, n [sgTTL] 168 h = 367 cells, n [sgTTL] 192 h = 386 cells, pool of three independent experiments. ****, P < 0.0001; **, P < 0.01; *, P < 0.05; ns, nonsignificant; logistic regression). **(F)** Protein lysates of control (sgControl) and TTL KO (sgTTL) U2OS cells were collected at different time points after lentiviral transduction and immunoblotted for cas9, Detyr-tub, TTL, Tyr-tub, and α-tubulin. GAPDH was used as loading control.

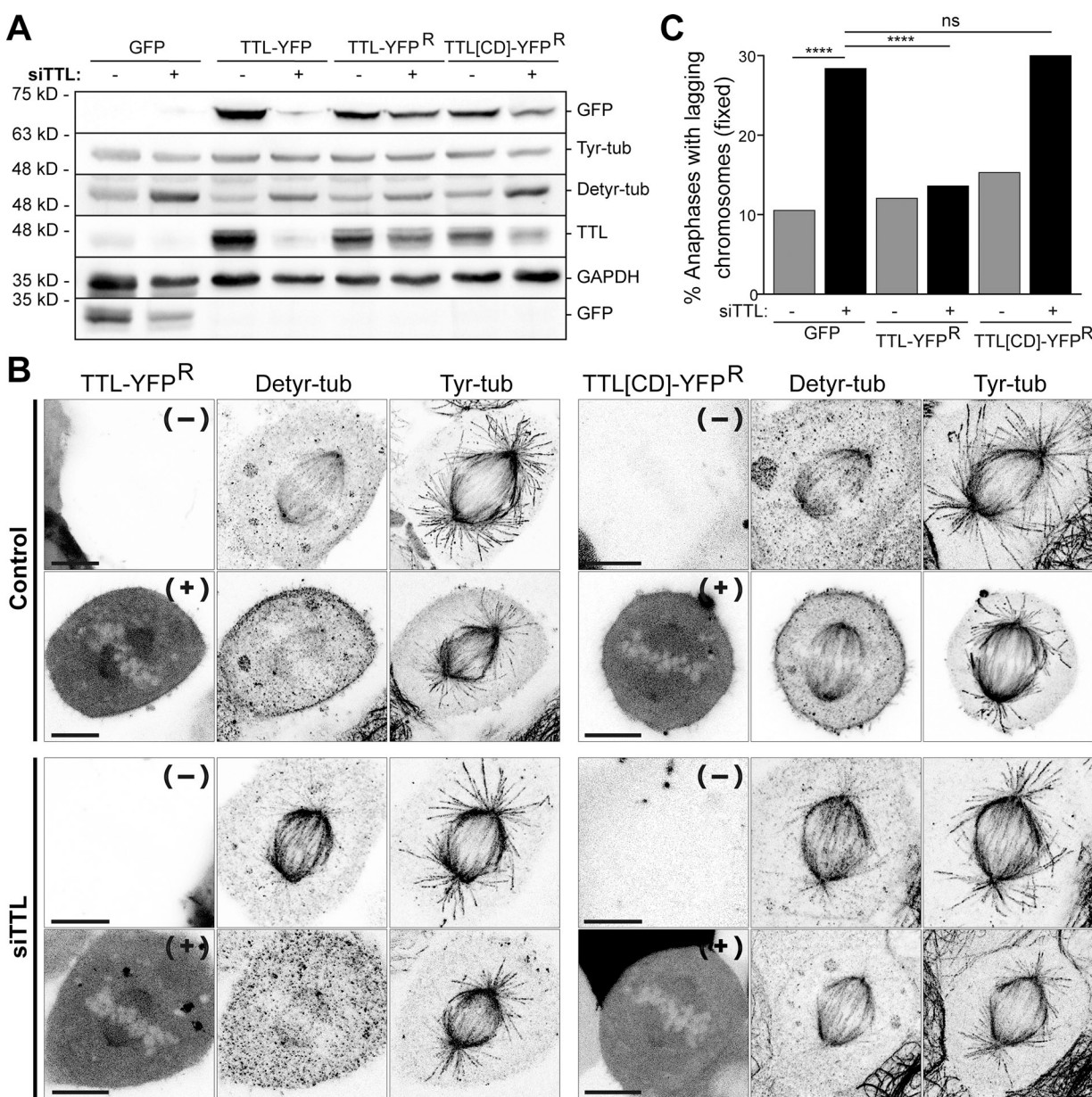

Figure S3.    **The catalytic activity of TTL is critical for α-tubulin retyrosination and prevents chromosome missegregation. (A)** Immunoblot analysis of tyrosinated and detyrosinated α-tubulin, TTL, GFP, and TTL-YFP levels after GFP, TTL-YFP, TTL-YFP$^R$, and TTL[CD]-YFP$^R$ exogenous expression. GAPDH was used as loading control. The expression of the TTL-YFP constructs was determined using an anti-GFP antibody. Representative immunoblot of three independent experiments. **(B)** Confocal analysis of U2OS cells expressing TTL-YFP$^R$ and TTL[CD]-YFP$^R$. Detyrosinated and tyrosinated α-tubulin and TTL-YFP$^R$/TTL[CD]-YFP$^R$ expression was analyzed by indirect immunofluorescence and by direct detection of the YFP signal, respectively. Maximum intensity 3D projection of representative cells for each condition. Scale bars, 10 µm. **(C)** Quantification of the percentage of anaphase cells with lagging chromosomes in control cells transiently expressing low-mild levels of GFP (control: 10.5% and siTTL: 28.4% [$n = 171$ and $n = 155$, respectively]), TTL-YFP$^R$ (control: 12.1% and siTTL: 13.6% [$n = 141$ and $n = 103$, respectively]), TTL[CD]-YFP$^R$ (control: 15.3% and siTTL; 30% [$n = 85$ and $n = 110$, respectively]). A pool of three independent experiments for each condition was used. ****, $P < 0.0001$, unpaired one-tailed $t$ test.

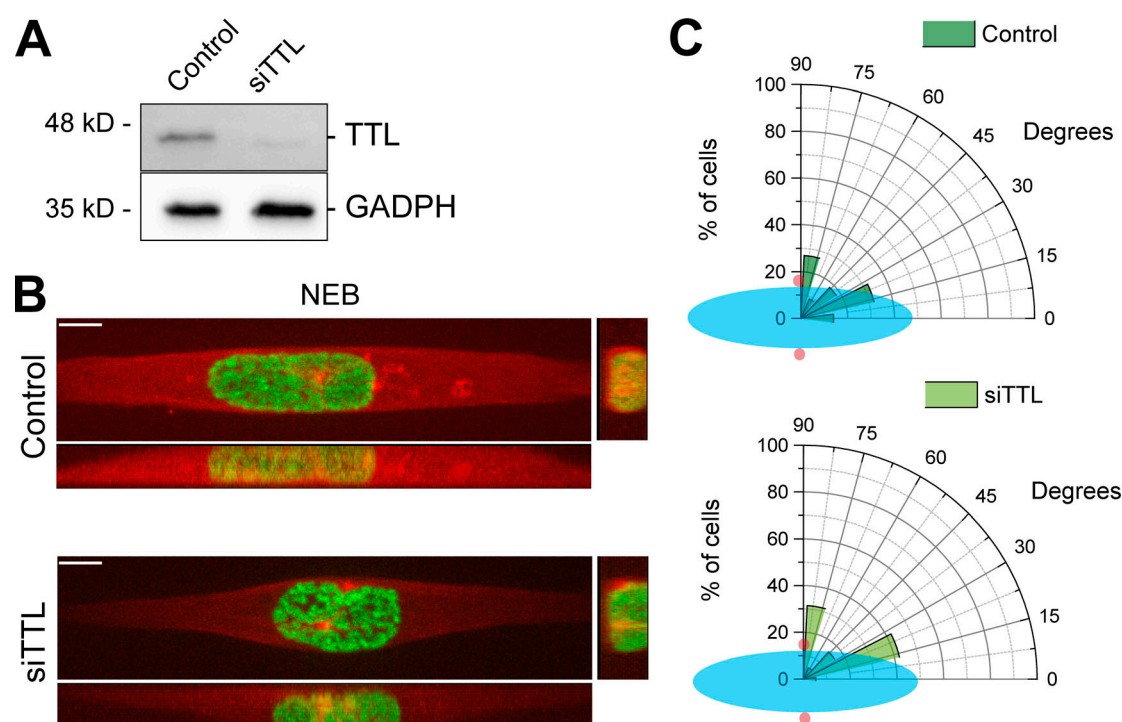

Figure S4. **The timing of centrosome separation at NEB is indistinguishable between control and TTL-depleted cells. (A)** Representative immunoblot to confirm TTL depletion efficiency by RNAi. **(B)** Live U2OS cells stably expressing H2B-GFP/mRFP-α-tubulin seeded on a horizontal, 10-µm-width line micropattern, at the moment of NEB, in control and after siTTL. Scale bars, 10 µm. **(C)** Polar plot showing centrosome positioning relative to the long nuclear axis at NEB for control and siTTL cells (n [control] = 22 cells, n [siTTL] = 19 cells, pool of three independent experiments, nonsignificant differences between conditions, nonparametric Kolmogorov–Smirnov test). Nuclear shape is represented by a blue ellipse, and centrosomes are represented by red circles.

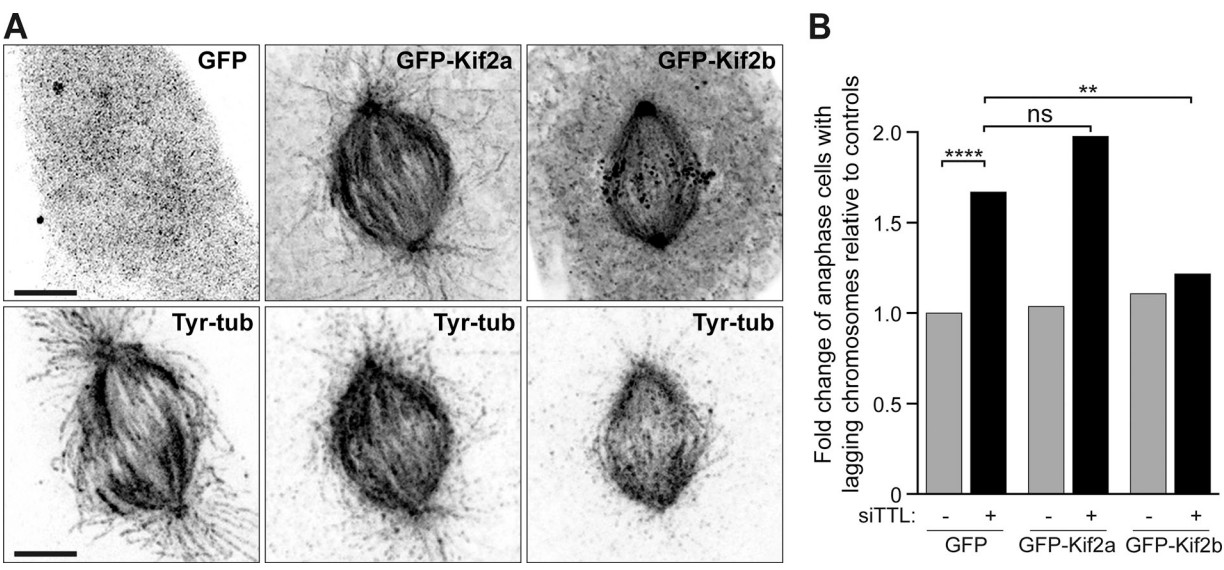

Figure S5. **Overexpression of the kinesin-13 Kif2b rescues chromosome missegregation due to increased α-tubulin detyrosination. (A)** 3D-deconvolution microscopy analysis of U2OS cells overexpressing GFP, EGFP-Kif2a, and EGFP-Kif2b and immunostained for tyrosinated α-tubulin. Scale bars, 5 µm. **(B)** Quantification of the fold change of anaphase cells with lagging chromosomes relative to controls (GFP) in rescue experiments using Kif2a and Kif2b overexpression. n (GFP) = 509, n (GFP+siTTL) = 530, n (Kif2a) = 296, n (Kif2a+siTTL) = 221, n (Kif2b) = 404, and n (Kif2b+siTTL) = 295, pool of two independent experiments. ns, nonsignificant; **, P < 0.01; ****, P < 0.0001; logistic regression.

