## [Peer Review File · The Journal of Cell Biology]

α -tubulin detyrosination impairs mitotic error correction by suppressing MCAK centromeric activity

Luisa Ferreira, Bernardo Orr, Girish Rajendraprasad, Antonio Pereira, Carolina Lemos, Joana Lima, Clàudia Guasch Boldú, Jorge Ferreira, Marin Barisic, and Helder Maiato

Corresponding Author(s): Helder Maiato, Institute for Molecular and Cell Biology University of Porto

Review Timeline:

Submission Date:	2019-10-11
Editorial Decision:	2019-11-24
Revision Received:	2019-12-30
Editorial Decision:	2020-02-04
Revision Received:	2020-02-04

Monitoring Editor: Hironori Funabiki

Scientific Editor: Marie Anne O'Donnell

Transaction Report:

DOI: <https://doi.org/10.1083/jcb.201910064>

November 24, 2019

Re: JCB manuscript #201910064

Prof. Helder Maiato
Institute for Molecular and Cell Biology University of Porto
Rua Alfredo Allen, 208
Porto 4200-135
Portugal

Dear Prof. Maiato,

Thank you for submitting your manuscript entitled "Microtubule tyrosination/detyrosination specifies a mitotic error code". The manuscript was assessed by expert reviewers, whose comments are appended to this letter. We invite you to submit a revision if you can address the reviewers' key concerns, as outlined here.

As you can see, both reviewers consider your work highly important, and I agree with this assessment, while reviewers and I find a few key issues to be addressed before publication in the JCB.

1. Rev#1 questions about your conclusion that microtubule "tyrosination/detyrosination works as a 'mitotic error code' that allows centromeric MCAK to discriminate correct and incorrect kinetochore-microtubule attachments", and I agree with this concern based on three points. First, this conclusion sounds like the detyrosination is first established to suppress MCAKdependent microtubule destabilization. However, as Rev#1 points out, since microtubule stabilization also promotes detyrosination, it is a feedback mechanism, and thus I feel that the statement is misleading. Second, from the term "mitotic error codes", naïve readers would assume a binary mode in which tyrosinated microtubules mark erroneous attachments, while detyrosinated microtubules mark correct attachments. However, this is clearly not the case, as demonstrated by increased merotelic attachments upon siTTL treatment. You may argue that the experiment illustrates an example of putting the "correct" code in a wrong context, but the fact that taxol enhances detyrosination rather than tyrosination fits well an idea that detyrosinated tubulins reflect stable microtubules rather than correct microtubule attachments. Third, in Abstract, you describe the phenotypic impacts of detyrosination and tyrosination, but your manipulation to control this processing is through indirect methods, such as depletion of TTL and V1/V2/SVBP. While you may have a choice to make this statement, considering the current science writing practice, I would suggest you describe the procedures to manipulate this processing in Abstract. In summary, I would like to ask you revise the title and the abstract through addressing these points.

2. Rev#2 asks you to test if siTTL phenotypes can be rescued by a catalytic mutant of TTL. In general, unless there is an issue of technical feasibility, it is a current standard in the cell biology field to assess the specificity of depletion/knockout phenotypes by complementation. Since the impact of detyrosination in chromosome segregation constitutes the major conclusion of the manuscript, this is a necessary additional experiment to be included in revision.

3. Reviewers also raise several other points, but I believe that you can address them without

additional experiments.

In addition, I would like to add some minor points to be revised:

A. Many phenotypic quantitations (e.g., Fig. 2f, l, 5c, 6d,e) combine data points from multiple experiments using logistic regression. I am concerned about this processing, as it is impossible for readers to assess data distribution, data variations among experiments and validate the statistic procedure. It would be best to present data so that these points can be assessed.

B. Fig.3e: please define the error bars.

C. The wording such as "microtubule detyrosination accumulates" (Abstract) sounds not right to me, as you are only measuring detyrosinated tubulins, but not detyrosination process. It would be better to use the term "detyrosinated tubulins" in this context.

D. Letters in Figure 7d are too small.

GENERAL GUIDELINES:

Text limits: Character count for an Article is < 40,000, not including spaces. Count includes title page, abstract, introduction, results, discussion, acknowledgments, and figure legends. Count does not include materials and methods, references, tables, or supplemental legends.

Figures: Articles may have up to 10 main text figures. Figures must be prepared according to the policies outlined in our Instructions to Authors, under Data Presentation, <http://jcb.rupress.org/site/misc/ifora.xhtml>. All figures in accepted manuscripts will be screened prior to publication.

*****IMPORTANT:** It is JCB policy that if requested, original data images must be made available. Failure to provide original images upon request will result in unavoidable delays in publication. Please ensure that you have access to all original microscopy and blot data images before submitting your revision.*******

Supplemental information: There are strict limits on the allowable amount of supplemental data. Articles may have up to 5 supplemental figures. Up to 10 supplemental videos or flash animations are allowed. A summary of all supplemental material should appear at the end of the Materials and methods section.

The typical timeframe for revisions is three months; if submitted within this timeframe, novelty will not be reassessed at the final decision. Please note that papers are generally considered through only one revision cycle, so any revised manuscript will likely be either accepted or rejected.

Thank you for this interesting contribution to Journal of Cell Biology. You can contact us at the journal office with any questions, cellbio@rockefeller.edu or call (212) 327-8588.

Sincerely,

Hironori Funabiki, Ph.D.
Monitoring Editor

Marie Anne O'Donnell, Ph.D.
Scientific Editor

Journal of Cell Biology

Reviewer #1 (Comments to the Authors (Required)):

This manuscript presents a systematic analysis of how changes in the tyrosination/detyrosination status of microtubules affects chromosome segregation. The experiments are thorough, carefully analyzed, and clearly presented. The results do provide a significant advance in our understanding of how this tubulin post-translational modification regulates mitosis, but I find the conclusions overstated at two levels.

First, the conclusion that "impaired error correction due to excessive microtubule detyrosination results from compromised MCAK activity" (heading, p. 9) is not well supported. The data do convincingly show that excessive detyrosination leads to errors and that increasing MCAK activity at centromeres reduces these errors. One explanation of these results is that high detyrosination compromises MCAK activity, which is the authors' interpretation. Another explanation is that detyrosination stabilizes kinetochore MTs by other mechanisms, and any perturbation that destabilizes would reduce errors. MCAK is one example, but another destabilizer (such as increased Aurora B activity or Ndc80 phosphorylation) could have a similar effect. Without an experiment to distinguish between these interpretations, I suggest that the conclusions be stated differently.

Second, the overall conclusion that tyrosination/detyrosination provides a "mitotic error code" is not well supported as currently presented. As I understand the model, it is essentially a positive feedback between stabilization and detyrosination: stable MTs are more detyrosinated, which further increases their stability. This process on its own cannot preferentially stabilize correct attachments because it relies on other mechanisms to initially discriminate and stabilize some subset of MTs. The stabilization/detyrosination cycle can reinforce other mechanisms (such as tension) that discriminate correct and incorrect attachments, and I suggest that the conclusion be stated in this way.

Minor comments:

Fig. 4a-d (top line of p. 10) should be Fig. 6?

Figure 5b should have some quantitation.

Reviewer #2 (Comments to the Authors (Required)):

Microtubules can be heavily post-translationally modified *in vivo*. The detyrosination/tyrosination has been shown to modulate the activity of MCAK, a microtubule depolymerase that plays an important role in mitotic error correction. Ferreira et al propose that the detyrosination state of the

microtubule plays a major role in mitotic error correction. They used a combination of state-of-the-art microscopy and gene editing techniques to understand the connection between microtubule tyrosination state, MCAK activity at the centromere and mitotic error correction. They show that detyrosination is associated with stable microtubules that have formed correct attachments to kinetochores. Mitotic errors due to excessive detyrosination are able to be rescued by MCAK centromeric activity and thus the tyrosination state of the microtubule plays an active role in mitotic correction machinery. This is an important *in vivo* finding for the functional consequences of microtubule posttranslational modifications. Overall the quality of the data is good and the figures are well presented and organized. The finding that MT modification state may play an active role in mitotic error correction is exciting. However, I have several concerns that I hope the authors can address in a revised manuscript:

- 1) For the data in Figure 1, the authors should show changes in tyrosination and detyrosination normalized to overall α -tubulin levels to be sure that detyr tubulin is increasing and not simply decreasing the tyr tubulin (while keeping the amount of detyr constant) It looks as if the taxol samples have less total tubulin (i.e. their fluoresce never gets above 50). Similarly for Figure 2. It looks like the siTTL has more total tubulin in the K-fibers than the control cells. This shows a nice correlation between siTTL and chromosome segregation, but could this be due to the presence of TTL and not its activity? Do the authors see the same results in a TTL depletion and expression of catalytically dead TTL? This an experiments that should be done.
- 2) The blue is very difficult to see in the 3-color overlay in figure 1. This should be changed to some other color that shows better contrast against the black background
- 3) Since the C-terminus of α -tubulin can also undergo Delta2 modification, the authors should test whether this modification is also enriched in the stabilized MTs with correct attachments.
- 4) In Figure 3, it would be good to see these cells fixed and stained for tyr/detyr tubulin as was done in fig. 1. While segregation errors increase in the siTTL case, it would be nice to see the localization of the tyr/detyr tubulin. I realize this is quantified in Suppl. Fig 1 but an image comparable to that shown in fig 2a would be good to have. Again, while the ratio of de-tyr/tyr is a good readout, I would also like to know what the total tubulin signal is.
- 5) Figure 4b shows a line connecting the data. Is this the line from the fit, or is it just connecting the data points? Given that the text states that the data can be fit well to a double exponential, why not extract the fraction of fast vs. slow turnover microtubules?
- 6) Is figure 5b similar to figure 2a ? The siTTL looks to have a lot more de-tyr tubulin (in figure 2a), or is this just a result of the look up table used in the image?
- 7) In the section titled "Impaired error correction due to excessive microtubule detyrosination" figures 4a-d are referenced. There is no figure 4d. Please check which figure is meant to be referenced here.
- 8) The data presented in Figure 6 nice data and a neat result. If detyrosination inhibits a restricted pool of centromeric MCAK, do you think that overexpressing TTL would rescue the % of lagging chromosome even more? The overexpression of GCPBM is nice, but to show that it is the microtubule modification state that is regulating this it would be nice to do this experiment in conditions in which the amount of tyrosinated MTs is increased by increasing TTL for example (or inhibiting VASH). In the gel in fig. 6b it looks like Δ NMCAK has more tyr tubulin than MCAKFL or

MCAKhyp while having the same amount of de-tyr tubulin. Is there an explanation for this?

9) This study defines a new role for spindle microtubules not as passive or structural elements of the spindle, but via their modifications as active players in the maintenance of correct MT-kinetochore attachments. The model is that as tension develops the correct attachments are stabilized and become detyrosinated which further stabilizes them. A discussion involving the rates of the enzymes responsible for detyrosination and tyrosination would be helpful. How long is needed to detyrosinate a microtubule? Is something else regulating the activity of the enzymes that detyrosinate these stable microtubules? These would be interesting points to add in the discussion.

Also, can the authors speculate on the evolutionary implications of this mechanism for organisms that do not have the tyrosination/detyrosination cycle?

Small point, but what is the evidence that MCAK is the "most potent microtubule depolymerase" in the Introduction. The study cited does not show a comparison between different kinesin 13 members.

Point-by-point response to reviewers and editor (in blue)

As you can see, both reviewers consider your work highly important, and I agree with this assessment, while reviewers and I find a few key issues to be addressed before publication in the JCB.

R: We thank the reviewers and editor for recognizing the importance of our findings. We hope to have addressed all the pending issues in this revised version.

1. Rev#1 questions about your conclusion that microtubule "tyrosination/detyrosination works as a 'mitotic error code' that allows centromeric MCAK to discriminate correct and incorrect kinetochore-microtubule attachments", and I agree with this concern based on three points. First, this conclusion sounds like the detyrosination is first established to suppress MCAK-dependent microtubule destabilization. However, as Rev#1 points out, since microtubule stabilization also promotes detyrosination, it is a feedback mechanism, and thus I feel that the statement is misleading. Second, from the term "mitotic error codes", naïve readers would assume a binary mode in which tyrosinated microtubules mark erroneous attachments, while detyrosinated microtubules mark correct attachments. However, this is clearly not the case, as demonstrated by increased merotelic attachments upon siTTL treatment. You may argue that the experiment illustrates an example of putting the "correct" code in a wrong context, but the fact that taxol enhances detyrosination rather than tyrosination fits well an idea that detyrosinated tubulins reflect stable microtubules rather than correct microtubule attachments. Third, in Abstract, you describe the phenotypic impacts of detyrosination and tyrosination, but your manipulation to control this processing is through indirect methods, such as depletion of TTL and V1/V2/SVBP. While you may have a choice to make this statement, considering the current science writing practice, I would suggest you describe the procedures to manipulate this processing in Abstract. In summary, I would like to ask you revise the title and the abstract through addressing these points.

R: These are all fair comments and we agree with all the points raised. We have therefore revised the title to better reflect our findings and explained in the abstract how experimental modulation of microtubule tyrosination/detyrosination was performed in cells.

2. Rev#2 asks you to test if siTTL phenotypes can be rescued by a catalytic mutant of TTL. In general, unless there is an issue of technical feasibility, it is a current standard in the cell biology field to assess the specificity of depletion/knockout phenotypes by complementation. Since the impact of detyrosination in chromosome segregation constitutes the major conclusion of the manuscript, this is a necessary additional experiment to be included in revision.

R: We have done the requested TTL RNAi rescue experiments either with RNAi-resistant WT-TTL or a catalytic dead mutant (E331Q; Szyk et al., Nat Struct Mol Biol, 2011; Prota et al., J Cell Biol, 2013). We found that RNAi-resistant WT-TTL, but not the corresponding catalytic dead version (without overexpression), was able to rescue the increased frequency of anaphase cells with lagging chromosomes observed upon TTL RNAi. These results not only demonstrate the specificity of the TTL RNAi phenotypes, but also directly indicate that the observed

missegregation events were dependent on the catalytic activity of TTL required for α -tubulin re-tyrosination. These data is now included in the manuscript as new Figure S3.

3. Reviewers also raise several other points, but I believe that you can address them without additional experiments.

R: We have addressed all the additional points raised by the reviewers (please see ahead).

In addition, I would like to add some minor points to be revised:

A. Many phenotypic quantitations (e.g., Fig. 2f, l, 5c, 6d,e) combine data points from multiple experiments using logistic regression. I am concerned about this processing, as it is impossible for readers to assess data distribution, data variations among experiments and validate the statistic procedure. It would be best to present data so that these points can be assessed.

R: We understand the expressed concerns because this is less common and more sophisticated statistics, but after consulting with a statistician expert, logistic regression was clearly indicated as the best option to assess statistical significance in some of our experiments. Namely, in those cases involving a very high number of independent experiments (up to 11 in some cases), high variability between experiments (despite the same trend), more than one variable per experiment (e.g. TTL RNAi + monastrol treatment + rescue with MCAK constructs), or a combination of these, logistic regression offers more powerful statistical validation. We also note that the data analysed in those experiments were categorized as discrete variables for the presence or absence of lagging chromosomes during anaphase. It is therefore a dichotomy (to have or not to have), rather than a broad distribution of data points. Our aim with logistic regression analysis was to assess whether any of the conditions would confer a higher or lower risk for chromosome missegregation. One of the advantages of logistic regression is that it takes into account the effect of a risk predictor (coefficient size) that evaluates the enhancement (positive coefficient) or reversal (negative coefficient) of a particular variable/phenotype (in our case, the presence of lagging chromosomes during anaphase). Finally, we note that data variation was also taken into account when we adjusted the logistic regression model for this variable, which is another advantage of this statistic procedure. This is now clarified in the methods section.

B. Fig.3e: please define the error bars.

R: These are now defined in the figure legend.

C. The wording such as "microtubule detyrosination accumulates" (Abstract) sounds not right to me, as you are only measuring detyrosinated tubulins, but not detyrosination process. It would be better to use the term "detyrosinated tubulins" in this context.

R: We have revised the terminology as suggested.

D. Letters in Figure 7d are too small.

R: We have increased the font size in Figure 7d.

Reviewer #1 (Comments to the Authors (Required)):

This manuscript presents a systematic analysis of how changes in the tyrosination/detyrosination status of microtubules affects chromosome segregation. The experiments are thorough, carefully analyzed, and clearly presented. The results do provide a significant advance in our understanding of how this tubulin post-translational modification regulates mitosis, but I find the conclusions overstated at two levels.

First, the conclusion that "impaired error correction due to excessive microtubule detyrosination results from compromised MCAK activity" (heading, p. 9) is not well supported. The data do convincingly show that excessive detyrosination leads to errors and that increasing MCAK activity at centromeres reduces these errors. One explanation of these results is that high detyrosination compromises MCAK activity, which is the authors' interpretation. Another explanation is that detyrosination stabilizes kinetochore MTs by other mechanisms, and any perturbation that destabilizes would reduce errors. MCAK is one example, but another destabilizer (such as increased Aurora B activity or Ndc80 phosphorylation) could have a similar effect. Without an experiment to distinguish between these interpretations, I suggest that the conclusions be stated differently.

R: We agree that any condition that destabilizes MTs overall would reduce errors and we now show that overexpression of ectopic Kif2b (a second microtubule depolymerase that localizes to prometaphase kinetochores), but not overexpression of ectopic Kif2a (a third microtubule depolymerase associated with microtubule minus-ends at spindle poles), is able to rescue the mitotic errors caused by excessive tubulin detyrosination after TTL RNAi (see new Figure S5). However, our measurements of kinetochore-microtubule half-life after TTL depletion indicate that the errors caused by increased detyrosination do not result from increased microtubule stability at kinetochores, as it would be required to support the alternative interpretation suggested by the reviewer. Importantly, driving constitutive localization of MCAK to centromeres (GCPBM) also rescued mitotic errors due to high tubulin detyrosination, and it did so without a significant alteration of overall kinetochore microtubule stability, as well as spindle and astral microtubule length. Taken together with published in vitro data showing that MCAK activity is strongly suppressed by tubulin detyrosination (Peris et al., J Cell Biol, 2009; Sirajuddin et al., Nature Cell Biol, 2014), these data favor our interpretation that mitotic errors due to excessive detyrosination do not result from global changes in kinetochore microtubule stability and rather reflect the ability of microtubule depolymerizing activities at centromeres/kinetochores to discriminate correct and incorrect attachments based on their detyrosination status. This is now discussed in detail in the main text.

Second, the overall conclusion that tyrosination/detyrosination provides a "mitotic error code" is not well supported as currently presented. As I understand the model, it is essentially a positive feedback between stabilization and detyrosination: stable MTs are more detyrosinated, which further increases their stability. This process on its own cannot preferentially stabilize correct attachments because it relies on other mechanisms to initially discriminate and stabilize some subset of MTs. The stabilization/detyrosination cycle can reinforce other mechanisms (such as tension) that discriminate correct and incorrect attachments, and I suggest that the conclusion be stated in this way.

R: We agree with the reviewer and have re-written the text to reflect the reviewer's suggestion.

Minor comments:

Fig. 4a-d (top line of p. 10) should be Fig. 6?

R: This is now corrected.

Figure 5b should have some quantitation.

R: Quantification of the % of spindles with high detyrosinated tubulin at the poles is now provided from three independent experiments.

Reviewer #2 (Comments to the Authors (Required)):

Microtubules can be heavily post-translationally modified in vivo. The detyrosination/tyrosination has been shown to modulate the activity of MCAK, a microtubule depolymerase that plays an important role in mitotic error correction. Ferreira et al. propose that the detyrosination state of the microtubule plays a major role in mitotic error correction. They used a combination of state-of-the-art microscopy and gene editing techniques to understand the connection between microtubule tyrosination state, MCAK activity at the centromere and mitotic error correction. They show that detyrosination is associated with stable microtubules that have formed correct attachments to kinetochores. Mitotic errors due to excessive detyrosination are able to be rescued by MCAK centromeric activity and thus the tyrosination state of the microtubule plays an active role in mitotic correction machinery. This is an important in vivo finding for the functional consequences of microtubule posttranslational modifications. Overall the quality of the data is good and the figures are well presented and organized. The finding that MT modification state may play an active role in mitotic error correction is exciting. However, I have several concerns that I hope the authors can address in a revised manuscript:

R: We thank the reviewer for sharing the enthusiasm with our findings. We hope now to have addressed all pending concerns.

1) For the data in Figure 1, the authors should show changes in tyrosination and detyrosination normalized to overall α -tubulin levels to be sure that detyr tubulin is increasing and not simply decreasing the tyr tubulin (while keeping the amount of detyr constant) It looks as if the taxol samples have less total tubulin (i.e. their fluoresce never gets above 50). Similarly for Figure 2. It looks like the siTTL has more total tubulin in the K-fibers than the control cells. This shows a nice correlation between siTTL and chromosome segregation, but could this be due to the presence of TTL and not its activity? Do the authors see the same results in a TTL depletion and expression of catalytically dead TTL? This an experiments that should be done.

R: We show that the levels of total tubulin (probing either with a pan α - or pan β -tubulin antibody) do not change significantly after TTL RNAi/KO or VASH1+SVBP overexpression (see western blots in figures 2D, 4A, 4C, 6B, S2A, S2C, S2F). In the particular case of human U2OS cells, tyrosinated tubulin comprises >90% of the total tubulin pool (Barisic et al., Science, 2015) and it is indistinguishable from total tubulin by immunofluorescence. We also note that

detyrosinated and tyrosinated tubulin are interconvertible and mutually exclusive. As so, an increase in detyrosinated tubulin is always accompanied with a decrease in tyrosinated tubulin (e.g. see figures 6B, S1A and S2F, where this is clearly perceptible). We also note that the immunofluorescence analysis was performed quantitatively keeping all acquisition parameters between different experimental conditions identical, and so even the absolute increase in detyrosinated tubulin, particularly in the TTL RNAi conditions in figure 2A, is evident. Lastly, our experiments involving fluorescence dissipation after photoactivation of GFP- α -tubulin revealed no detectable changes in the proportion of kinetochore microtubules, as well as kinetochore microtubule half-life between control and TTL-depleted cells, further supporting that total tubulin is essentially unaltered. Therefore, quantifying the frequency of detyrosinated tubulin as a ratio relative to tyrosinated tubulin along individual k-fibers or astral microtubules provides a robust measure of the relative proportion of detyrosinated vs. tyrosinated tubulin in the different experimental conditions, independently of inherent differences in spindle architecture.

Regarding the last points, we have performed the requested TTL RNAi rescue experiments either with RNAi-resistant WT-TTL or a catalytic dead mutant (E331Q; Szyk et al., Nat Struct Mol Biol, 2011; Prota et al., J Cell Biol, 2013). We found that RNAi-resistant WT-TTL, but not the corresponding catalytic dead (without overexpression), was able to rescue the increased frequency of anaphase cells with lagging chromosomes observed upon TTL RNAi. These results not only demonstrate the specificity of the TTL RNAi phenotypes, but also directly indicate that the observed missegregation events were dependent on the catalytic activity of TTL required for α -tubulin re-tyrosination. These data is now included in the manuscript as new Figure S3.

2) The blue is very difficult to see in the 3-color overlay in figure 1. This should be changed to some other color that shows better contrast against the black background

R: We have adjusted the blue contrast in the merged image to better reflect what is reported in the inverted contrast images of single channels.

3) Since the C-terminus of α -tubulin can also undergo Delta2 modification, the authors should test whether this modification is also enriched in the stabilized MTs with correct attachments.

R: This is an interesting suggestion but Delta2 modification is very low abundant and only reliably detectable with an antibody by Western Blot with a high protein load.

4) In Figure 3, it would be good to see these cells fixed and stained for tyr/detyr tubulin as was done in fig. 1. While segregation errors increase in the siTTL case, it would be nice to see the localization of the tyr/detyr tubulin. I realize this is quantified in Suppl. Fig 1 but an image comparable to that shown in fig 2a would be good to have. Again, while the ratio of de-tyr/tyr is a good readout, I would also like to know what the total tubulin signal is.

R: We now provide immunofluorescence analysis showing tyr/detyr tubulin in the requested experiments.

5) Figure 4b shows a line connecting the data. Is this the line from the fit, or is it just connecting the data points? Given that the text states that the data can be fit well to a double exponential, why not extract the fraction of fast vs. slow turnover microtubules?

R: The double exponential fitting is done for each experiment, but the lines shown in Figure 4b (and Figure 7b) simply connect the data points representing the mean \pm SD from the different experiments. Otherwise we would be doing a curve fit from different curve fits, with no real meaning. This is now clarified in the methods and main text. We also extracted the average fraction of fast vs. slow turnover microtubules for each experimental condition and included these data in new Figure 4 along with proper discussion in the main text.

6) Is figure 5b similar to figure 2a ? The siTTL looks to have a lot more de-tyr tubulin (in figure 2a), or is this just a result of the look up table used in the image?

R: The techniques used in those figures were different. Figure 2a was obtained using CH-STED, whereas old Figure 5B was obtained using wide-field + 3D-deconvolution. All other figures used conventional scanning confocal microscopy. Note also that different fixation protocols were used to preserve astral MTs (glutaraldehyde+paraformaldehyde) in Figure 2a, as opposed to more standard fixation with ice cold methanol in old Figure 5b. To harmonize conditions between figures, we now provide a new panel obtained using scanning confocal microscopy included in new Figure 5.

7) In the section titled "Impaired error correction due to excessive microtubule detyrosination" figures 4a-d are referenced. There is no figure 4d. Please check which figure is meant to be referenced here.

R: This is now corrected.

8) The data presented in Figure 6 nice data and a neat result. If detyrosination inhibits a restricted pool of centromeric MCAK, do you think that overexpressing TTL would rescue the % of lagging chromosome even more? The overexpression of GCPBM is nice, but to show that it is the microtubule modification state that is regulating this it would be nice to do this experiment in conditions in which the amount of tyrosinated MTs is increased by increasing TTL for example (or inhibiting VASH). In the gel in fig. 6b it looks like Δ NMCAK has more tyr tubulin than MCAKFL or MCAKhyp while having the same amount of de-tyr tubulin. Is there an explanation for this?

R: This would be a great experiment indeed, however the problem with TTL overexpression is that it strongly compromises CENP-E-mediated chromosome congression (Barisic et al., Science, 2015), precluding any subsequent analysis of anaphase events. TTL depletion is a much more benign condition as most cells resume congression and enter anaphase only with a slight delay. The requested experiment is therefore not feasible. Nevertheless, we draw attention to Figure 3, where we show that depletion of VASH1+2 (with or without SVBP depletion), which only partially decreases detyrosinated tubulin (and consequently increases tyrosinated tubulin), rescues the % of lagging chromosomes even beyond control levels. Regarding the minor differences in the amount of tyr tubulin in the different MCAK mutants, these are likely due to slight variability in the expression of the different MCAK mutants and/or in the amount of loaded protein.

9) This study defines a new role for spindle microtubules not as passive or structural elements of the spindle, but via their modifications as active players in the maintenance of correct MT-kinetochore attachments. The model is that as tension develops the correct attachments are

stabilized and become detyrosinated which further stabilizes them. A discussion involving the rates of the enzymes responsible for detyrosination and tyrosination would be helpful. How long is needed to detyrosinate a microtubule? Is something else regulating the activity of the enzymes that detyrosinate these stable microtubules? These would be interesting points to add in the discussion.

R: These points are now discussed in the revised manuscript.

Also, can the authors speculate on the evolutionary implications of this mechanism for organisms that do not have the tyrosination/detyrosination cycle?

R: This speculation is now included in the revised manuscript.

Small point, but what is the evidence that MCAK is the "most potent microtubule depolymerase" in the Introduction. The study cited does not show a comparison between different kinesin 13 members.

R: We agree with the reviewer and have now re-written the problematic sentence.

February 4, 2020

RE: JCB Manuscript #201910064R

Prof. Helder Maiato
Institute for Molecular and Cell Biology University of Porto
Rua Alfredo Allen, 208
Porto 4200-135
Portugal

Dear Dr. Maiato:

Thank you for submitting your revised manuscript entitled " α -tubulin detyrosination impairs mitotic error correction by suppressing MCAK centromeric activity". Both reviewers are supportive of publication, and I agree with this assessment. However, I found that my previous minor point C, related to the wording like "detyrosination accumulates" still remains to be addressed. These should be corrected to be "detyrosinated α -tubulins", since it is not accurate to state that enzymatic reactions accumulate. These examples are:

Abstract: "Here we found that α -tubulin detyrosination accumulates on correct, more stable, kinetochore-microtubule attachments."

Page 5. " α -tubulin detyrosination accumulates on correct, more stable, kinetochore-microtubule attachments".

In addition, reasoning behind the word "Paradoxically" in Abstract is confusing to me, since in mitosis field, it is accepted that kinetochore microtubule attachment is stabilized at the metaphase plate, but stabilization interferes with error-correction. So, I would suggest you edit this sentence.

Please be sure to address any remaining concerns in the final version of the manuscript. Pending these revisions and any revisions necessary to meet our length and other formatting guidelines (see details below), we would be happy to publish the paper in JCB.

- Add a paragraph after the Materials and Methods section briefly summarizing the online supplementary materials

A. MANUSCRIPT ORGANIZATION AND FORMATTING:

Full guidelines are available on our Instructions for Authors page, <http://jcb.rupress.org/submission-guidelines#revised>. **Submission of a paper that does not conform to JCB guidelines will delay the acceptance of your manuscript.**

B. FINAL FILES:

-- High-resolution figure and video files: See our detailed guidelines for preparing your production-ready images, <http://jcb.rupress.org/fig-vid-guidelines>.

Thank you for this interesting contribution, we look forward to publishing your paper in Journal of Cell Biology.

Sincerely,

Hironori Funabiki, Ph.D
Monitoring Editor

Marie Anne O'Donnell, Ph.D.
Scientific Editor

Journal of Cell Biology

Reviewer #1 (Comments to the Authors (Required)):

The authors have satisfied my concerns, and I recommend publication.

Reviewer #2 (Comments to the Authors (Required)):

The authors have addressed all my comments. This is a thorough and elegant study and should be published in JCB.

I just have a minor comment that I hope the authors will consider.

In the Discussion when mentioning the preference of TTL for soluble tubulin, the correct references would be Arce et al. 1975 and Raybin and Flavin 1975 and not Ersfeld et al 1993 which reports the cloning of TTL.